# Extracellular ATP/P2X7 receptor, a regulatory axis of migration in ovarian carcinoma-derived cells

José David Nuñez-Ríos[1], Mauricio Reyna-Jeldes[2,3,4], Esperanza Mata-Martínez[1], Anaí del Rocío Campos-Contreras[1¤], Iván Lazcano-Sánchez[1], Adriana González-Gallardo[1], Mauricio Díaz-Muñoz[1], Claudio Coddou[2,3,4], Francisco G. Vázquez-Cuevas[1]*

1 Departamento de Neurobiología Celular y Molecular, Instituto de Neurobiología, Universidad Nacional Autónoma de México, Querétaro, México, 2 Departamento de Ciencias Biomédicas, Facultad de Medicina, Universidad Católica del Norte, Coquimbo, Chile, 3 Millennium Nucleus for the Study of Pain (MiNuSPain), Santiago, Chile, 4 Núcleo Para el Estudio del Cáncer a Nivel Básico, Aplicado y Clínico, Universidad Católica del Norte, Coquimbo, Chile

¤ Current address: Facultad de Ingeniería, Universidad Autónoma de Querétaro, Querétaro, México
* fvazquez@comunidad.unam.mx

**Data Availability Statement:** Microarray data were deposited by pur group in ArrayExpress. Accesion number:E-MTAB-13598. All other relevant data are

## Abstract

ATP is actively maintained at high concentrations in cancerous tissues, where it promotes a malignant phenotype through P2 receptors. In this study, we first evaluated the effect of extracellular ATP depletion with apyrase in SKOV-3, a cell line derived from metastatic ovarian carcinoma. We observed a decrease in cell migration and an increase in transepithelial electrical resistance and cell markers, suggesting a role in maintaining a mesenchymal phenotype. To identify the P2 receptor that mediated the effects of ATP, we compared the transcript levels of some P2 receptors and found that *P2RX7* is three-fold higher in SKOV-3 cells than in a healthy cell line, namely HOSE6-3 (from human ovarian surface epithelium). Through bioinformatic analysis, we identified a higher expression of the *P2RX7* transcript in metastatic tissues than in primary tumors; thus, P2X7 seems to be a promising effector for the malignant phenotype. Subsequently, we demonstrated the presence and functionality of the P2X7 receptor in SKOV-3 cells and showed through pharmacological approaches that its activity promotes cell migration and contributes to maintaining a mesenchymal phenotype. P2X7 activation using BzATP increased cell migration and abolished E-cadherin expression. On the other hand, a series of P2X7 receptor antagonists (A438079, BBG and OxATP) decreased cell migration. We used a CRISPR-based knock-out system directed to *P2RX7*. According to the results of our wound-healing assay, SKOV3-P2X7^KO cells lacked receptor-mediated calcium mobilization and decreased migration. Altogether, these data let us propose that P2X7 receptor is a regulator for cancer cell migration and thus a potential drug target.

## Introduction

Ovarian cancer is the most lethal gynecological cancer. According to GLOBOCAN, the worldwide lethality rate for ovarian cancer was 68% in 2020, compared to 30% for breast cancer,

within the manuscript and its Supporting information files.

**Funding:** This study was supported by: Dirección General de Asuntos del Personal Académico (DGAPA), Universidad Nacional Autónoma de México, grant IN205223 to F.G.V.-C. Fondo Nacional de Desarrollo Científico y Tecnológico (FONDECYT grant 1161490), Comisión Nacional de Investigación Científica y Tecnológica (FONDEQUIP grant EQM140100), and MiNuSPain (Chile) to C.C. Funders had no role in study design, data collection and analysis, decision to publish, or preparation of this article.

**Competing interests:** The authors have declared that no competing interests exist.

making ovarian cancer a significant health problem that must be addressed [1]. Epithelial cancer (carcinoma) is the most frequent type of cancer in the ovary. It is characterized by a high invasive potential, which causes it to spread to other organs, such as the uterus, the liver and the brain [2, 3]. In addition to metastasis based on the intravasation of tumor cells into lymphatic and blood vessels, ovarian carcinoma (OvCar) has a particular mechanism of spread known as transcoelomic spread, which involves the acquisition of a metastatic phenotype throughout a trans-differentiation process called epithelial-to-mesenchymal transition (EMT). During EMT, cancer cells migrate from the primary tumor to the peritoneal cavity, where they are propelled by ascitic fluid dynamics to other organs and develop secondary tumors [4, 5]. These characteristics make OvCar highly aggressive. Therefore, understanding the mechanisms that regulate the ability to migrate and induce a mesenchymal phenotype is essential to overcoming this disease.

ATP is present at nanomole levels in healthy tissue but has been shown to be a significant factor in the tumor microenvironment, where concentrations can reach hundreds of micromoles [6]. High concentrations of ATP in cancerous tissues can activate any purinergic receptor and consequently regulate multiple cellular processes. As analyzed in recent reviews, ATP and its metabolites (e.g., adenosine) regulate physiological processes in tumor cells, such as cell proliferation and migration, in an autocrine-paracrine manner. Similarly, the purinergic system orchestrates the antitumoral immune response, commanding immune cell recruitment and reprogramming to favor tumor growth [7–10].

Aside from its important role in metabolism, ATP acts as an intercellular messenger by activating P2 receptors, which are made up of two subfamilies: P2YR and P2XR. The former belongs to the G-protein-coupled receptors superfamily and the latter is a ligand-activated ion channel. Seven genes encoding for seven P2X subunits have been described so far. Each P2X subunit has two transmembrane domains and a long extracellular domain, with the corresponding amino and carboxy ends intracellularly located. Functional receptors can be either homo- or heterotrimers of these subunits. P2X receptors mediate $Na^+$ and $Ca^{2+}$ influx and $K^+$ efflux but differ in their desensitization kinetics [11].

One member of the P2X receptor subfamily gained special attention due to its characteristics. P2X7 is a homotrimeric receptor that does not desensitize in the presence of the agonist. Moreover, in its fully activated state, it can open the so-called "megapore," a conductance that allows the passage of molecules of around 900 Da, thus inducing apoptosis by $Ca^{2+}$ overload [11]. On the other hand, its constitutive basal activity promotes aerobic glycolysis by maintaining an intracellular $Ca^{2+}$ tone [12, 13]. Furthermore, P2X7 receptor has a long carboxy end that contains multiple consensus sites for protein-protein interactions that are important in signal transduction such as a consensus site for Src tyrosine kinase and for β-arrestin scaffold protein [14], both of which regulate cell survival and proliferation.

Characterizations of P2X7 receptor expression in the cancer context are controversial. It acts as a tumor suppressor in endometrial [15], uterine [16], cervical [17], esophageal squamous cell [18], and urogenital and distal paramesonephric epithelial cancers [19], since its expression is reduced compared to P2X7 receptors in healthy tissue. In other cancers, such as prostate [20], thyroid [21], breast [22] and ovarian [23], P2X7 becomes highly expressed with respect to healthy tissue, suggesting a pro-tumoral role.

Evidence has also shown that P2X7 can induce apoptosis in some carcinoma cells [24, 25]. However, recent findings suggest that overexpression of the P2X7 receptor in non-tumorigenic HEK293 embryonic kidney cells supports tumor progression by transforming the cells and inducing the ability to develop tumors [26]. Furthermore, it has been described that the P2X7 receptor supports cell proliferation and survival in carcinoma models [23, 27] in addition to cell migration and metastatic spreading [28, 29].

The opposing actions of the P2X7 receptor can be explained by the occurrence of splicing variants of the *P2RX7* transcript. Research has shown that full P2X7 receptor activation can induce pore formation and promote cell death [30], whereas variants like P2X7B [31] or P2X7J [17] cannot. Regarding these splicing variants, P2X7B has gained attention in cancer research because it contributes to tumor progression, metastasis, and resistance to chemotherapy [32].

In previous work, our group demonstrated that P2X7 receptor expression is higher in ovarian carcinoma compared to non-cancerous tissue and that inhibiting the receptor with the selective antagonist A438079 reduces AKT (Ser473) and ERK (Thr202/Tyr204) phosphorylation, resulting in a decrease in cell viability and proliferation rate, which contributes to cancer progression [23].

In the present work, we analyzed the effect of extracellular ATP hydrolysis with the nucleotidase apyrase (Apy) on the migration of metastatic (SKOV-3) and non-metastatic (CAOV-3) ovarian carcinoma-derived cells. We determined that extracellular ATP supports cell migration in metastatic cells. By comparing the relative expression of purinergic receptors with reverse transcription followed of quantitative polymerase chain reaction (RT-qPCR) between SKOV-3 cells and non-tumorous human ovarian surface epithelial (HOSE) cells, we found that P2X7 was overexpressed in metastatic SKOV-3 cells, making it a promising mediator for the actions of extracellular ATP on cell migration. Thus, using pharmacological approaches and knocking out the P2X7 receptor, we concluded that P2X7 mediates extracellular ATP actions by maintaining a constant rate of migration and invasion. Altogether, our data demonstrate that the extracellular ATP/P2X7 axis supports the mesenchymal phenotype in metastatic ovarian carcinoma-derived cells.

## Materials and methods

### Cell characteristic and culture

SKOV-3 (ATCC ID HTB-77) cell line is a model of metastatic serous ovarian carcinoma, it was obtained from the ascites from a white 64 years old woman, being hypodiploid with 43 chromosomes in approximately 63% of the cells, this line carries mutations in TP53 and in the subunit alpha of PIK3CA, a detailed description of mutations in genes of importance for cancer is available [33]. CAOV-3 (ATCC HTB-75) cells were isolated from a primary ovarian adenocarcinoma of a white 54-year-old patient.

SKOV-3 and CAOV-3 cell lines were obtained from ATCC and were grown in RPMI-1640 and Dulbecco's Modified Eagle Medium (DMEM), respectively, under standard cell culture conditions (37°C, 5% of $CO_2$ and relative humidity). Cell culture medium was supplemented with 10% FBS and 1% antibiotic/antimycotic (both from Gibco). All cell cultures were allowed to reach 60–80% confluence unless stated otherwise.

### Publicly available data

We used Gene Expression Omnibus DataSets to localize RNA-seq and microarray data about metastatic ovarian cancer. We identified three studies that used tumor samples from metastatic tissues and primary tumors from the same patient to compare metastatic and primary tumor conditions. Each study was analyzed using the GEO2R tool [34]. We set the Benjamini —Hochberg (false discovery rate) option for P-value adjustment and graphed Log2 fold change values.

In addition, we used the Kaplan-Meier plotter [35] to compare high and low expression levels of P2RX7 by evaluating changes in serous and endometroid histology at each stage of the disease.

For this project we used data published by our group [36] (Accession number E-MTAB-13598). Briefly, SKOV-3 cell lines were stimulated with Apy (10 u/mL) for 24 h using SKOV-3 cells with no treatment as a control. Three different plates with each stimulus were collected and submitted for microarray analysis. Gene expression differences are presented as Z-score means. Genes that were differentially expressed, both up and down, and had an absolute value greater than two were used. This gene list was analyzed using ShinyGO v0.77 [37] and the REACTOME (version 86) platform [38].

## qPCR experiments

Minimum information for publication of quantitative real time polymerase chain reaction experiments (MIQE) guidelines were considered in every aspect of the following methodology in order to ensure its replicability, robustness, and reproducibility [39]. The human ovarian adenocarcinoma cell line SKOV-3 and its non-tumoral ovarian surface epithelial counterpart HOSE6-3 were seeded on 60 mm culture dishes at a proportion of 50,000 cells/dish to achieve 60–80% confluence after incubating cells for 24 h at 37˚C + 5.0% $CO_2$. mRNA was extracted using TRIzol reagent (Invitrogen, Waltham, MA, USA) according to the manufacturer's guidelines. Yield and purity ratios (260 nm/280 nm and 260 nm/230 nm) were established for each sample using a NanoDrop One® microvolume spectrophotometer (Thermo Fisher Scientific, Waltham, MA, USA). Reverse transcription was performed using the Affinity Script qPCR cDNA Synthesis Kit (Agilent Technologies, Santa Clara, CA, USA), employing 2.0 μg of extracted mRNA and considering a final reaction volume of 20 μL, as well as including a no retrotranscriptase control (-RT) for each sample. For qPCR reactions, Brilliant II SYBR® Green QPCR Master Mix (Agilent Technologies, Santa Clara, CA, USA) was used for 100 ng of cDNA and 300 nM as a final primer concentration for each gene. For relative expression assessment, P2X7R, P2Y2, P2Y4R, P2Y6R and P2Y11R were defined as target genes, and B2M (Beta-2-Microglobulin) was used as a referential gene. Primer Blast was used for primer design to obtain an amplicon size between 50 and 250 bp and an annealing/extension temperature of 60˚C. Efficiency curves were generated for all primer pairs to select a proper relative expression quantification method (Table 1). The thermal protocol used for all experiments started with a denaturation step at 95˚C for 10 min, followed by an amplification phase of 40 cycles consisting of 30 s at 95˚C and 60 s at 60˚C. Fluorescence was detected at the end of each cycle. After amplification, melting curves were performed on the amplified products, incubating them at 95˚C for 60 s, ramping down to 55˚C and then increasing temperature to 95˚C at a rate of 0.2˚C/s while continuously measuring fluorescence data. As an additional identity control, 10 μL of each amplified sample was used for amplicon size detection with GelRed 1X (Biotium, Fremont, CA, USA)-stained 1.5% agarose gels. To calculate relative gene expression, the Livak and Schimittgen method was applied [40]. For all of these experiments, analysis was performed using the HOSE6-3 condition as the basal expression control, and statistical comparisons were made between HOSE6-3 cells and their corresponding SKOV-3 cell line.

## Transepithelial electrical resistance measurement

SKOV-3 cells were seeded in 12-well Transwell plates with 8.0 μm pore size inserts (Corning, Corning, NY, USA), considering an initial seeding of 150,000 cells/well to achieve 60–80% confluence after a 24 h incubation at 37˚C + 5.0% $CO_2$. After this incubation, SKOV-3 conditions were established, treating these cells with 10 U/mL Apy (Sigma-Aldrich, St. Louis, MO, USA), 100 μM ATP (Sigma-Aldrich, St. Louis, MO, USA), 100 nM A438079 (Tocris Bioscience, Bristol, UK), and 10 minute-boiled Apy solutions in culture medium for 48 h. After these treatments, transepithelial electrical resistance (TEER) measurements were performed

**Table 1. Primer sequences and parameters.**

| Gene | NCBI Reference Sequence | Primer Sequence | Amplicon size (bp) | Amplicon melting T° (°C) | Primer Efficiency (%) |
|---|---|---|---|---|---|
| hP2X7 | NM_002562.6 | Fw: 5'-TGCACACCAAGGTGAAGGGGA-3' | 195 | 81.0 ± 0.5 | 105 |
| | | Rv: 5'-TGCGGGTGGGATACTCGGGA-3' | | | |
| hP2Y2 | NM_176072.3 | Fw: 5'-CAGGTCCAGGCGTGTGCATT-3' | 150 | 84.0 ± 0.5 | 107 |
| | | Rv: 5'-AGCCACCTGACCAGGGCTTT-3' | | | |
| hP2Y4 | NM_002565.4 | Fw: 5'-CCGGGAAGTGAGAGAAAAGGGGATG-3' | 94 | 79.0 ± 0.5 | 107 |
| | | Rv: 5'-AGCCTGGAAAAGAGGAAGAAGCACC-3' | | | |
| hP2Y6 | NM_176796.3 | Fw: 5'-CAGAACATTGCACGCGACAGTTTCA-3' | 229 | 83.5 ± 0.5 | 103 |
| | | Rv: 5'-GTGGGTTTCCTATGTTCAGGGAGGC-3' | | | |
| hP2Y11 | NM_002566.5 | Fw: 5'-ACAGGACTGGAGACGCAAGAACAAA-3' | 180 | 85.5 ± 0.5 | 101 |
| | | Rv: 5'-CTTATACCTGCCACCCTCCCCTACC-3' | | | |
| hB2M | NM_004048.4 | Fw: 5'-AAGTGGGATCGAGACATGTAAGCA-3' | 70 | 77.0 ± 0.5 | 106 |
| | | Rv: 5'-GGAATTCATCCAATCCAAATGCGGC-3' | | | |

daily for 5 days using an EVOM2 Epithelial Voltohmmeter with an STX2 electrode (World Precision Instruments, Sarasota, FL, USA). TEER was calculated according to Kim et al. [41], considering a growth area of 1.12 $cm^2$ and the untreated SKOV-3 cell condition as the control situation for statistical analysis.

## Stress fiber counting

SKOV-3 cells were cultured on cover slips when 75–80% confluence was reached. Apy (10 u/mL) was applied and cultures were incubated for 24 h. Afterwards, cells were dyed with rhodamine-coupled phalloidin (Thermo Fisher, Waltham, MA, USA) according to the manufacturer's instructions. Pictures were acquired by confocal microscopy (Zeiss LSM 780) and analyzed using ImageJ (NIH, Bethesda, MD, USA). The color channels were split and only the red channel was used. Afterwards, a straight line was drawn from one side of the cell membrane to the other, and increments in fluorescence were plotted. Finally, peaks higher than 50% of the maximal signal were selected and counted, correlating this number to the number of stress fibers. This process was done three times for each cell. Fifty cells were used for each treatment.

## Pharmacological stimulation

Cells were cultured in 35 mm plates and allowed to reach 70% confluence. Once ready, cells were starved in RPMI medium without serum. Afterwards, a fresh medium containing 1% of fetal bovine serum (FBS) was added with its corresponding treatment: control with no drug, 2′(3′)-O-(4-Benzoylbenzoyl)adenosine-5′-triphosphate BzATP 50 μM, A438079 125 nM, Brilliant blue G (BBG) 200 nM and oxidized ATP (OxATP) 200 μM. After incubating for 24 h, cells were homogenized using Laemmli 2x buffer, and western blotting was performed as previously described [23]. We used an antibodies designed to recognize vimentin (Cell Signaling 3932), E-cadherin (Cell Signaling 144725) and β-actin (Sigma A2066) as housekeeping. Each experiment was performed in triplicate.

## Isolation of membranal proteins by biotinylation

Plasmatic membrane proteins exposed on the surface of cells were biotinylated by incubating with 300 μM of the membrane impermeable reactive Ez-link Sulfo-NHS-LC-Biotin (Thermo

Scientific, USA) by 30 minutes, in phosphate buffer (PBS: in mM136.8 NaCl, 26,8 KCl, 10,1 Na2HPO4, 1.76 KH2PO4, pH 7.4). Then, cells were lysed in TNTE buffer (in mM: 50 Tris-HCl pH 7.4, 150 NaCl, 1 EDTA, and 0.1% Triton X-100), insoluble bodies discarded and protein concentration determined by Lowry method. 200 μg of total protein were incubated with 10 μL of the affinity avidin-sepharose beads (Cell Signaling Technology, USA) for 120 min. After this time, beads were washed by 3 times and prepared to electrophoresis. P2X7 receptor was detected by western blot with two antibodies, one directed against COOH-end (Alomone, Jerusalen, Israel, #APR-004) of and other against the extracellular loop of the receptor (Alomone, Jerusalen, Israel, #APR-008).

## Calcium imaging recording

$Ca^{2+}$ recording was done according to previously published methods [42]. Briefly, cells were cultured in coverslips for 48 h, then they were incubated for 25 min with 2 μM Fluo-4AM (Thermo Fisher Scientific, Waltman, MA, USA) in Krebs buffer (in mM 150 NaCl, 1 KCl, 1,5 CaCl2, 1 MgCl2, 10 HEPES and 4 glucose; pH 7,4) at 37°C in an incubator containing an atmosphere of 95% $O_2$, 5% $CO_2$. Then, cultures were washed to remove the not-incorporated Fluo-4AM and placed in a recording chamber coupled to a Ts2R-FL fluorescence microscope (Nikon, Minato, Tokio, Japan) where the stimulus was manually applied. Videos were acquired at 500 msec of frequency with a Retiga Electro CCD camera drive using Ocular software (Teledyne photometrics, Tucson, AR, USA) and a 20X objective. Extracellular solutions used in the recordings were: N-$Ca^{2+}$: [$Ca^{2+}$] = 1.5 mM or Z-$Ca^{2+}$ (without $Ca^{2+}$ and with 2 mM EGTA). For the latter, the [$Ca^{2+}$] calculated with Maxchelator (Webmaxc standard UC Davis) was = $2.4X10^{-11}$ M.

To normalize the data, 10 μM of ionomycin and 5 mM of $MnCl_2$ were added at the end of each recording. Raw data were analyzed with ImageJ software (NIH, Bethesda, MD, USA). Normalization was done with the equation (F/F0)-1(100))/(Fmax), where F = fluorescence value in a given time, F0 = Basal fluorescence value without any stimulus and Fmax = maximum value of fluorescence obtained after ionomycin addition.

## Wound-healing assay

SKOV-3 and CAOV-3 cells were seeded in 35 mm culture plates and maintained until reaching total confluence under standard conditions (37°C, 5% $CO_2$ and relative humidity) and monolayers were formed. Next, the cells were starved for 8 h using culture media with no FBS. The monolayer was scratched with a yellow pipette tip to make a wound, followed by washing three times with PBS. After the final wash, fresh medium was added with 1% of FBS and the following experimental conditions: control with no treatment, BzATP 50 μM, A438079 125 nM, BBG 200 nM and OxATP 200 μM. Five pictures were taken for each experiment, and every experimental condition was evaluated in triplicate. Pictures were taken at times 0 h and 16 h. The wound area was measured using ImageJ and setting time 0 h as the total wound area and the difference between 0 h and 16 h as the wound closure. Data are presented as percentages and normalized to control conditions.

## Zebrafish xenograft assay

The zebrafish xenograft trial is a procedure designed to evaluate the viability and migration of tumor cells, it involves a small number of experimental subjects in a completely controlled environment, the cells are transplanted through a single injection and the time to observe the results is short, which implies a short time of animal stress (48h). This study was carried out in strict accordance with the national and international guidelines for the care and use of

laboratory animals. The protocol was approved by the Ethics for Research Committee of the Instituto de Neurobiología. Wild-type zebrafish embryos were obtained from crosses of adult male and female fishes. Embryos (n = 10) were collected at the one cell stage and kept under standard conditions at 28°C in E3 medium (5 mM NaCl, 0.17 mM KCl, 0.33 mM CaCl2 and 0.33 mM MgSO4), with 14:10 h light-dark cycles, when they reached 48 h post fertilization were dechorionated and immersed in E3 medium containing 0.04 mg/ml tricaine (Sigma-Aldrich). After 7–10 minutes, when larvae zebrafish lost the righting reflex, approximately 200 cells were microinjected into the perivitelline space. The injected cell mass was confirmed visualizing the perivitelline cavity, this moment was considered time 0, injected fishes were transferred to E3 medium at 33°C for the next 48 h. Then, fishes were anesthetized and observed by epifluorescence microscopy, after which the animals were immediately euthanized by anesthesia overdose, as no further experimental procedures were performed on them.

For this experiment, SKOV-3-GFP+ cells were used to facilitate their tracking. Prior injection, cells were incubated for 2 h with either OxATP 200 μM in DMEM-F12 or DMEM-F12 media alone (control). Pictures were taken at times 0 h, and 48 h for each zebrafish. Tumor size was measured drawing a straight line from one side of the cell to the other. Three measurements per embryo were obtained and averaged. Each experiment was made in triplicate.

## Results

### 1.- Depleting extracellular ATP reverts the mesenchymal phenotype of OvCar cell lines

Our group has previously demonstrated that SKOV-3 cells can release ATP to the extracellular media and that ATP depletion using Apy can modify gene expression and inhibit migration [36]. In that study, our group gathered microarray data after incubating SKOV-3 cells with Apy (10 U/mL by 12 h) and observed the cell processes affected by such treatment. For this study, our goal was to analyze the genes that were significantly changed after Apy treatment. To do so, we utilized ShinyGO 0.77, a robust and curated platform [37]. Additionally, we aimed to confirm whether such genes are involved in migration-related pathways. We entered the list of genes that had a significant z-score in either up- or down-regulation, and many of the genes we saw modified in our microarray are involved in migration-related pathways such as Hippo, JAK-STAT, Rap1, WNT and Notch (Table 2). In order to draw firm conclusions about the microarray data, we entered the gene list into the REACTOME (version 86) platform [38] and observed involvement of those genes in Reactome cell migration pathways (Table 3).

This analysis suggests that ATP has a role in cell migration. To confirm this role and understand how extracellular ATP supports the mesenchymal characteristics displayed by SKOV-3 cells, we incubated SKOV-3 cultures with Apy (10 U/mL) and evaluated the migration ability, trans-epithelial electrical resistance (TEER) and expression level of EMT markers (vimentin and E-cadherin).

To evaluate the effect on cell migration, we used wound-healing and Boyden chamber assays. Incubating SKOV-3 cells with Apy (10 U/mL) for 16 h reduced basal migration in the scratch assay to 69.32 ± 1.85% (data normalized to control, / 1A). With the transwell assay, we identified a 46.86 ± 11.6% (normalized to control, S1 Fig) decrease in cell migration. To investigate if this decrease was linked to actin-cytoskeleton arrangements, we evaluated the number of stress fibers in response to incubation with Apy using the line profile tool in Image-J as described in Methods. A significant decrease in fibers was detected for Apy-treated cells (5.36 ± 0.24 and 2.3 ± 0.15 fibers per line for control and Apy, respectively) (Fig 1B). Then, expression levels of phenotypic markers (epithelial or mesenchymal) were evaluated by Western blotting. E-cadherin levels rose noticeably (0.97 ± 0.66 and 7.48 ± 0.93 for control and Apy

**Table 2. Enriched pathways for microarray data.**

| Fold Enrichment | Pathway |
|:---:|:---|
| 1.76 | Hippo signaling pathway |
| 1.68 | Rap1 signaling pathway |
| 1.66 | Pathways in cancer |
| 1.48 | JAK-STAT pathway |
| 1.48 | MTOR signaling pathway |
| 1.46 | PI3K-Akt signaling pathway |
| 1.45 | Ras signaling pathway |
| 1.44 | Wnt signaling pathway |
| 1.42 | Notch signaling pathway |

respectively), while vimentin levels remained relatively unchanged (1.14 ± 0.41 and 0.96 ± 0.20), (Fig 1C). To confirm whether these observations were an effect of ATP depletion or just a characteristic displayed by the SKOV-3 cell line, we performed a scratch assay on the non-metastatic ovarian carcinoma cell line CAOV-3. We found that after incubating with Apy, CAOV-3 cells diminished cell migration (55.44 ± 8.35%, data normalized to control), reinforcing the idea that ATP depletion modulates cell migration in a negative manner; furthermore, expression of P2RX7 transcript was confirmed (S3 Fig).

In addition, TEER was evaluated as an indicator of epithelial or mesenchymal-like phenotypes. This assay reflects the degree of cell coupling in a monolayer. Thus, SKOV-3 cells were cultured for 5 days under control conditions or were incubated with Apy (10 U/mL), and TEER was measured daily. TEER values increased from day 1 in Apy-treated cells (35.7±0.85 and 53.2±0.75 $\Omega cm^2$ for control and Apy-treated cells respectively). Cells were kept for 5 days, and a significant difference was observed over the course of the experiment (50.7±0.75 versus 70.7±2.0 $\Omega cm^2$ for control and Apy, respectively, on day 2; 60.2±1.37 versus 78.5±1.93 $\Omega cm^2$ on day 3; 62±1.4 versus 98.5±1.1 $\Omega cm^2$ on day 4; and 65.5±0.9 versus 103.0±3.2 $\Omega cm^2$ on day 5). A group of cells was treated with boiled Apy as a control. Under this condition no significant changes were observed compared to the control group (Fig 1D).

Such effects orchestrated by Apy-mediated ATP hydrolysis suggested that extracellular ATP supports a mesenchymal-like and more migrative phenotype in ovarian carcinoma cells.

## 2.- P2X7 receptor is overexpressed in ovarian-derived cancer cell lines

Given that incubating with Apy significantly changed the phenotype of SKOV-3 cells (favoring a more epithelial one), we inferred that the Apy effects might be explained by the abrogation of a receptor-dependent signaling pathway resulting from the hydrolysis of purinergic ligands by

**Table 3. Enriched REACTOME pathways for microarray data.**

| Pathway name | Description |
|:---:|:---:|
| RHOV GTPase cycle | Atypical RHO GTPase that is thought to be constitutively active in cancer; it might be involved in WNT and JNK pathways and regulate cell adhesions |
| RHOJ GTPase cycle | Regulates the cytoskeleton, including formation of lamellipodia and actin filaments; highly expressed in endothelial cells and regulates their motility and vascular morphogenesis |
| Activation of RAC1 | Drives Actin polymerization and formation of lamellipodia, promotes cell-cell adhesion and breakdown and migration of different cancer cells |

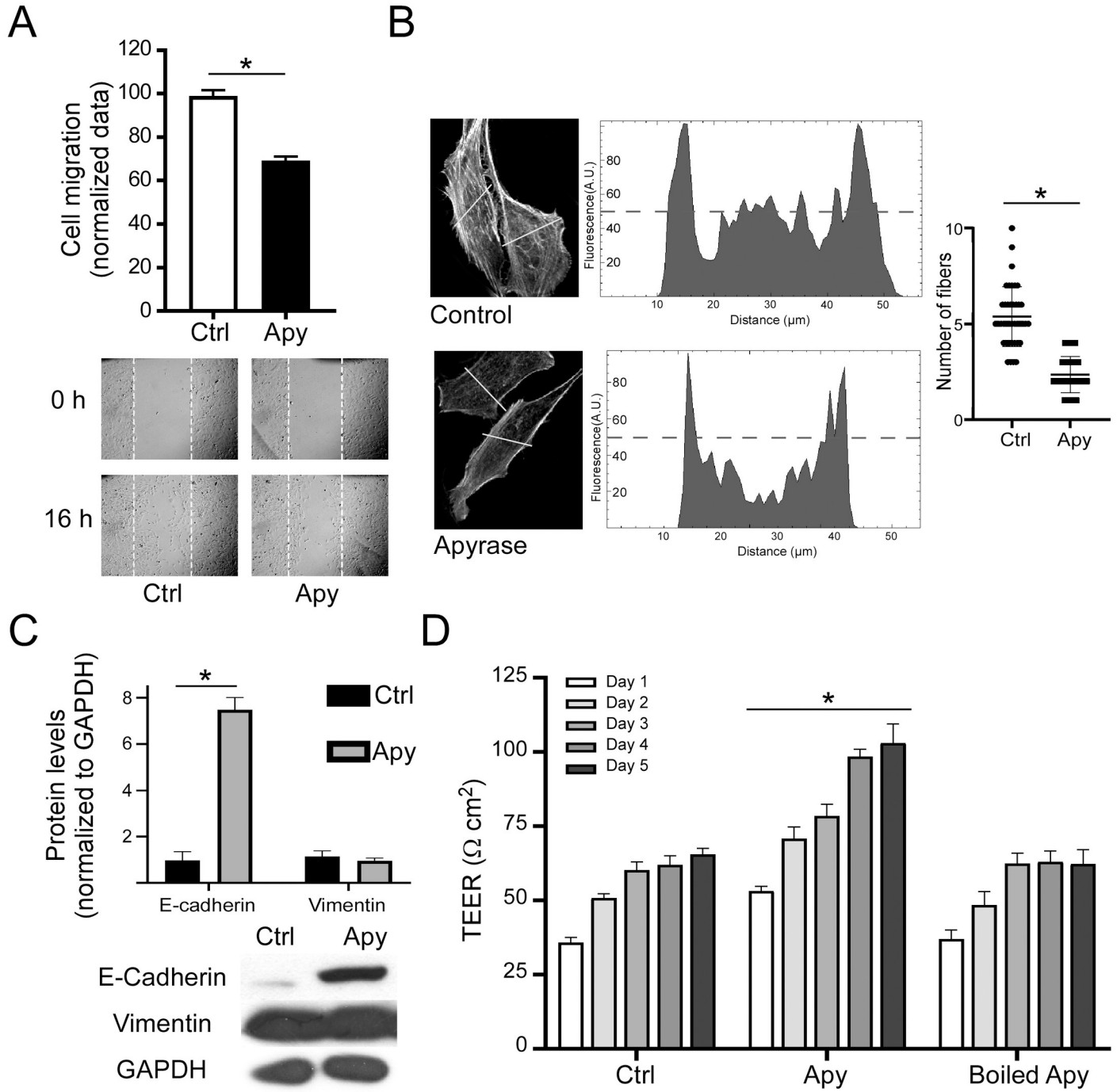

**Fig 1. Depleting extracellular ATP reverts the mesenchymal phenotype of ovarian carcinoma cell lines.** A) Confluent cultures of SKOV-3 cells were incubated for 16 h with 10 U/mL of Apy. Afterwards, a wound-healing assay was performed. The upper panel shows the mean ± S.E.M. of three independent experiments, *p<0.05 Student´s t-test. The lower panel shows representative pictures. B) Line profile analysis in SKOV-3 cells treated with 10 U/mL of Apy for 16 h stained for F-actin with phalloidin-rhodamine. Confocal pictures on the left show representative cells. The line used for the analysis is drawn in yellow. At the center, histograms show the fluorescence level in arbitrary units (A.U.), relative to the distance of the drawn line (in μm). The graph on the right shows the mean ± S.E.M. of N cells from three independent experiments, *p<0.05 Student´s t-test. C) TEER measurement in SKOV-3 cells treated with 10 U/mL of Apy for 5 days; determinations were made each day. As negative control, the nucleotidase was inactivated (boiled Apy). The graph shows the mean ± S.E.M. from three independent experiments, *p<0.05 ANOVA. D) Western blot detection of epithelial-mesenchymal markers E-cadherin and vimentin. The graph shows the mean ± S.E.M. of 4 independent experiments, *p<0.05 Student´s t-test. The lower panel shows representative pictures.

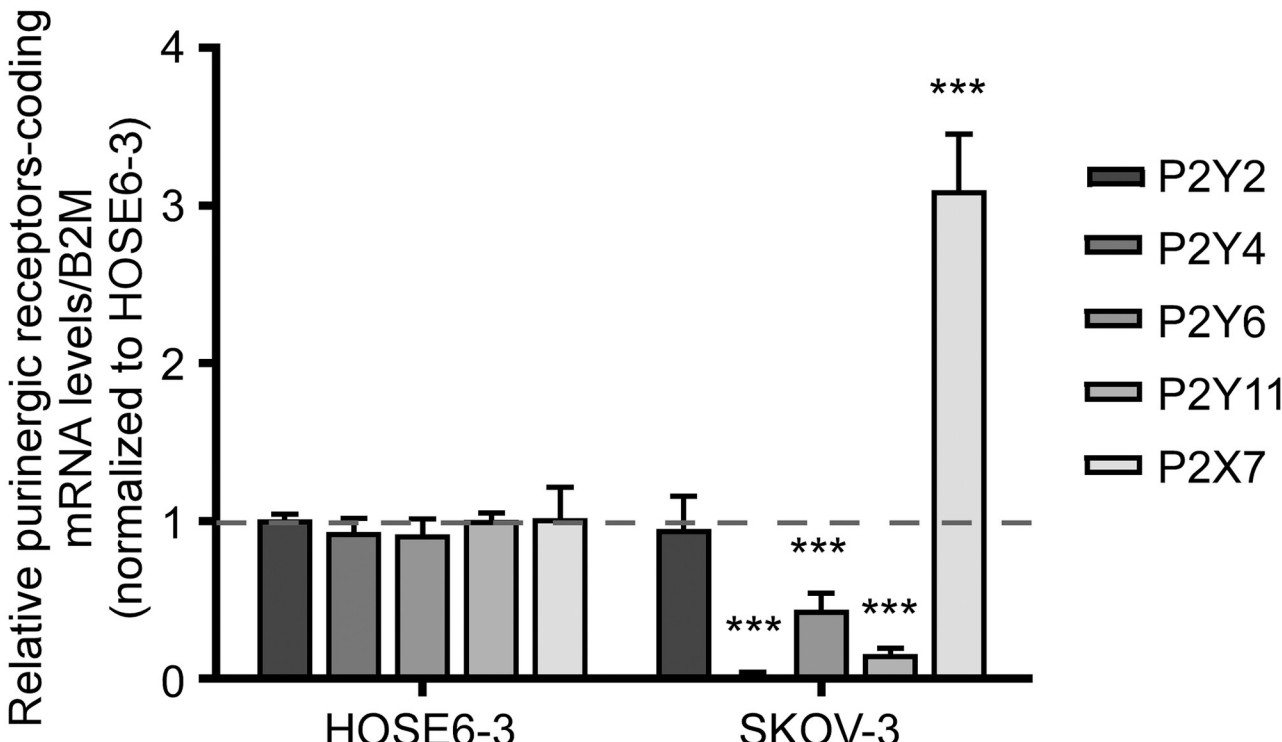

**Fig 2. P2X7 receptor is overexpressed in ovarian-derived cancer cell lines.** Relative expression evaluated using real-time PCR and the $2^{-\Delta\Delta CT}$ method for transcripts coding for P2Y2, P2Y4, P2Y6, P2Y11 and P2X7 receptors in SKOV-3 cells, normalized against HOSE6-3 levels. The housekeeping transcript B2M (beta-2-microglobulin) was used. The graph shows the mean ± S.E.M. of 4 independent experiments ***p<0.01 Student´s-t test.

the nucleotidase. We compared the expression levels of purinergic receptor mRNAs in SKOV-3 cells to the non-transformed ovarian surface epithelium cell line HOSE6-3 to identify the purinergic receptor responsible for inducing mesenchymal-like traits (Fig 2). Through qPCR, we determined that the *P2RX7* transcript is the only purinoceptor overexpressed in SKOV-3 cells (3.1±0.17-fold, data normalized to HOSE6-3 values), suggesting that P2X7 receptor is the main mediator of the extracellular ATP effects that are abrogated by Apy. Based on these observations, we investigated the role of P2X7 receptor in the regulation of ovarian carcinoma cell phenotype.

### 3.- P2X7 is overexpressed in OvCar biopsies and is related to a worse diagnosis and metastasis

To investigate the possible relevance of P2X7 receptor in ovarian carcinoma, we analyzed public databases. First, we consulted the *human protein atlas* [43] to verify the expression of P2X7 in samples with different histology. Our search yielded 29 ovarian carcinoma biopsies that covered different histological subtypes (15 serous, 7 mucinous and 6 endometroid), and 87% of the samples were positive to P2X7 receptor. A representative picture is shown in Fig 3A.

After confirming the expression of P2X7 in different OvCar samples, we used public transcriptomic data to identify a relationship between receptor expression and clinical outcome. Specifically, we employed the Kaplan-Meier Database platform [35] and looked for a relationship between the *P2RX7* transcript levels and the progression-free survival (PFS) of OvCar patients. For this, cohorts of patients with high transcription levels were compared to those

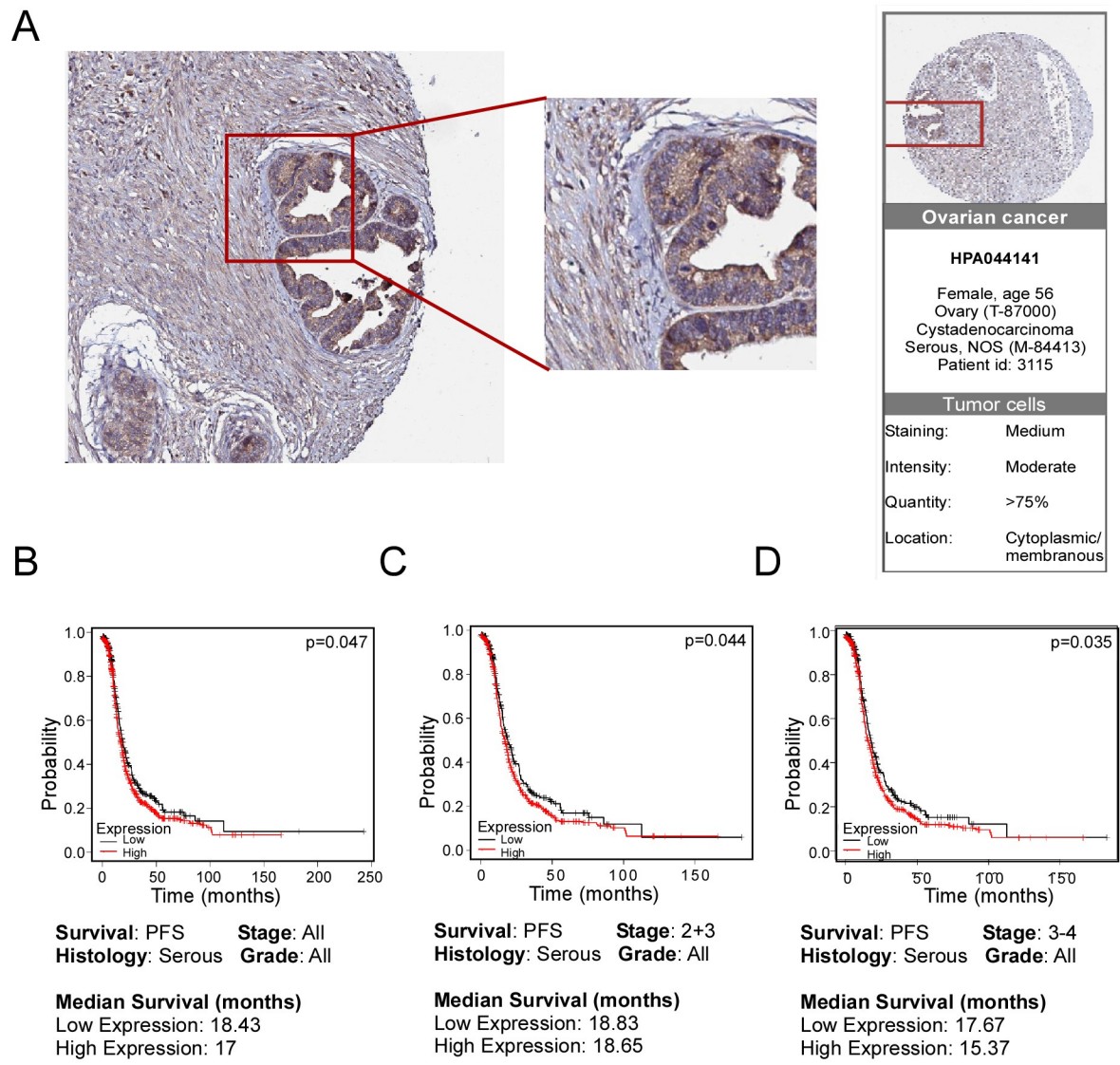

**Fig 3. P2X7 is overexpressed in ovarian cancer biopsies and is related to a worse diagnosis and metastasis.** A) Representative picture of P2X7 receptor expression in a serous carcinoma biopsy, from *human protein atlas*. The positive signal is in brown and the tissue stain is in blue. The tissues were analyzed with the validated antibody HPA044141. B-D) Relationship between the expression level of the *P2RX7* transcript and the probability of survival of patients with ovarian carcinoma created with data from the Kaplan-Meier Plotter Database. Patients are classified as follows: B) Serous ovarian carcinoma patients (361 low expression and 743 high expression) C) Serous ovarian carcinoma patients stages 2+3 (294 low expression and 609 high expression) and C) Serous ovarian carcinoma patients stage 3+4 (333 low expression and 668 high expression).

with low transcription levels. We observed that patients with high transcription levels of *P2RX7* exhibited a lower PFS for serous histopathology in all stages (p = 0.047; Fig 3B), serous histopathology in stage 2+3 (p = 0.044, Fig 3C) and serous histopathology in stage 3+4 (p = 0.035; Fig 3D).

Given that P2X7 is correlated with a poor prognosis, we wanted to explore this notion in comparison with other transcripts coding for purinergic receptors, emulating the qPCR analysis for the metastatic SKOV-3 cells and non-transformed HOSE6-3 cells shown in Fig 2. Thus, we surveyed transcriptomic data from GEO and found three different data sets from which we

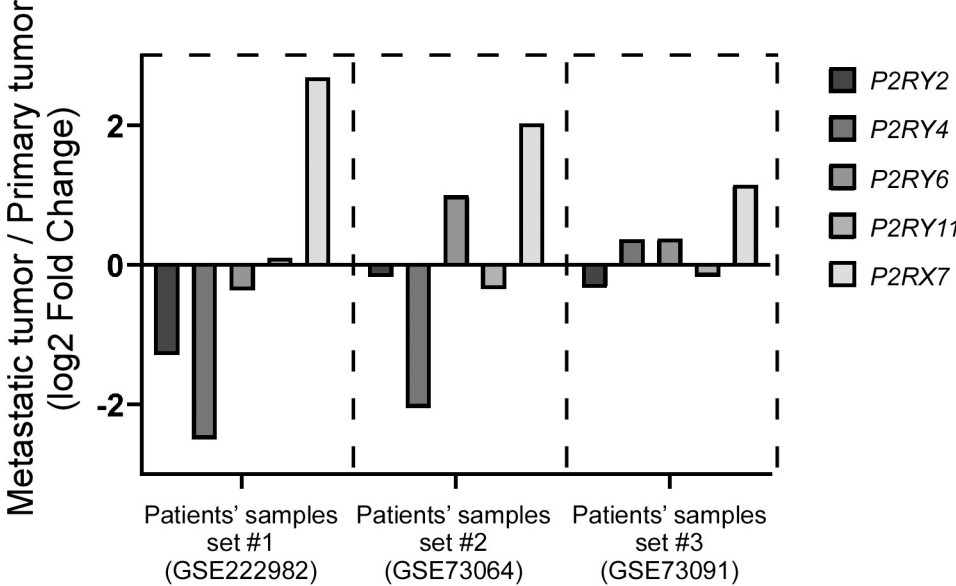

**Fig 4. P2RX7 is overexpressed in metastatic ovarian carcinoma tumors.** Transcription levels of purinergic receptors were analyzed and compared between metastatic and primary tumors, and plotted in means of Log2FoldChange. Each metastatic sample was compared to its corresponding primary tumor from the same patient. Corresponding accession numbers: (Patient samples 1: GSE222982, n = 5), (Patient samples 2: GSE73064, n = 5), (Patient samples 3: GSE73091, n = 3).

could compare a metastatic tumor to a primary ovarian tumor in the same patient (Accession numbers: GSE222982 [44], GSE73064 [45] and GSE73091 [46]). We entered these data sets into GEO2R and searched for the Log2 Fold Change (Log2FC) values in the *P2RY2*, *P2RY4*, *P2RY6*, *P2RY11* and *P2RX7* transcripts (S1 Table). Only the *P2RX7* Log2FC value increased (1.15, 2.03 and 2.69, in the three data sets) when comparing metastatic tissue to primary tumor tissue (S1 Table). The other purinergic receptor transcripts hardly increased, and some were even downregulated (-0.33, -0.17 and -1.29 for *P2RY2*; 0.37, -2.05 and -2.29 for *P2RY4*; 0.38, 0.99 and -0.37 for *P2RY6*; -0.17, -0.35 and 0.11 for *P2RY11*) (Fig 4). Higher transcription levels of *P2RX7* in metastatic tissue compared to primary tumor tissue suggest that P2X7 plays a role in OvCar metastasis.

We can thus conclude that P2X7 receptor is undoubtedly expressed in ovarian carcinoma tissue and that it is important to characterize its function in cancer cell physiology.

## 4.-Ovarian carcinoma cells express functional P2X7 receptor

To demonstrate the presence and functionality of the P2X7 receptor we first employed RT-PCR to detect the *P2RX7* transcript, a pair of oligonucleotides was designed against the 5′ end of P2RX7 transcript (NM_002562.2), the primer forward sequence is locate into the exon 2 and the reverse primer in the junction of exons 3 and 4 of the P2RX7 locus (ENSG00000089041); thus, the primer design is unable to amplify from genomic DNA. It was obtaining a 195 bp amplicon that was sequenced and identified in the BLAST platform, matching human P2RX7 (Fig 5A). To observe the P2X7 protein expressed in SKOV-3 cells, we made western blot using an antibody directed against the COOH-end and a second antibody directed against the extracellular loop, in a preparation enriched with plasmatic membrane

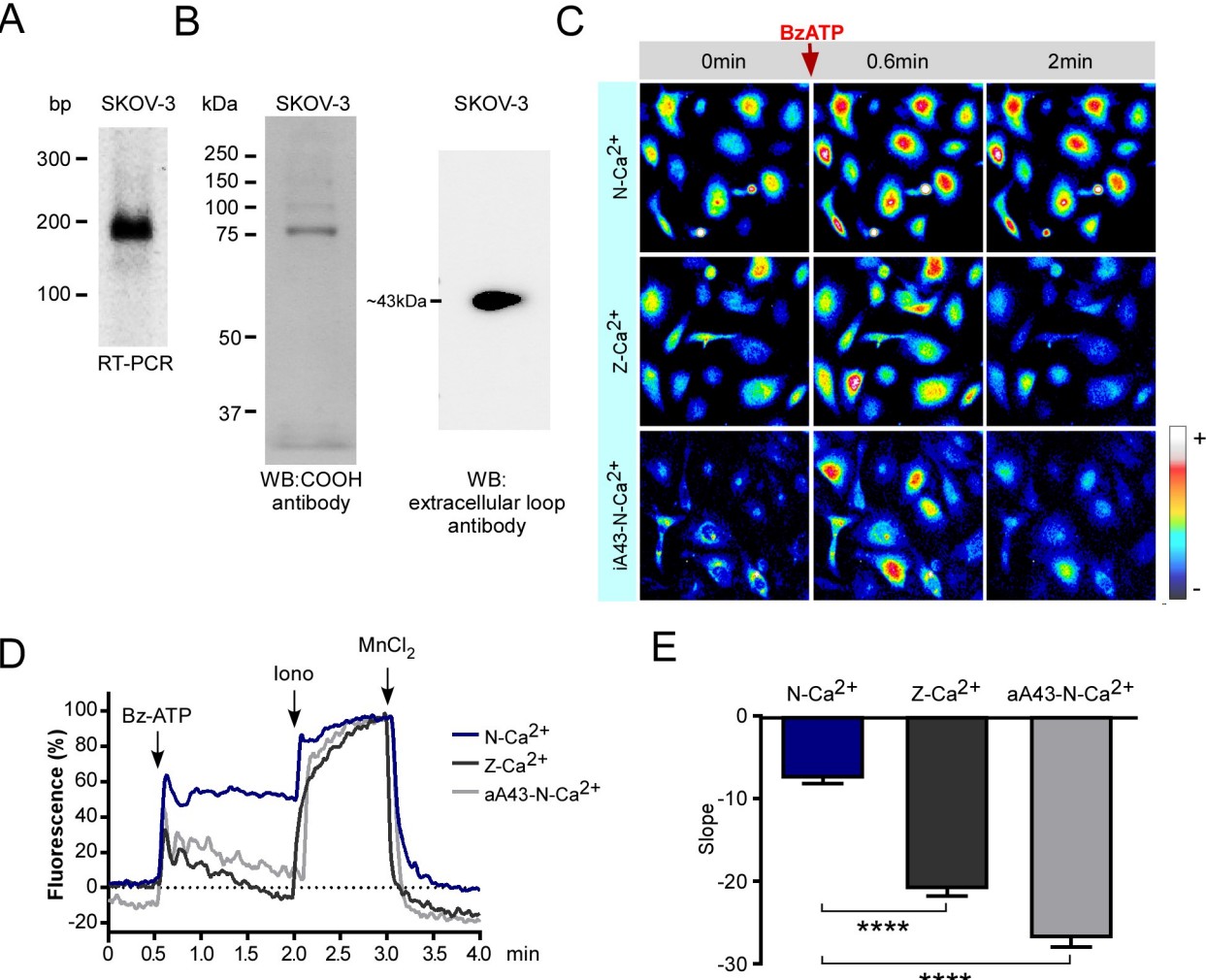

**Fig 5. P2X7 is functionally expressed in the ovarian carcinoma cell line SKOV-3.** A) The P2RX7 transcript was analyzed by reverse transcription followed by end-point PCR. Amplicons were analyzed in 1.5% agarose gels, purified and sequenced to confirm their identity. B) Western blot detection of P2X7 receptor in a preparation enriched in plasmatic membrane proteins from SKOV-3 cells, the protein was detected with an antibody directed against COOH-end (left) and other to the extracellular loop (right) C) Sequence of fluorescent images from SKOV-3 cells loaded with Fluo4-AM 2 μM showing changes in $[Ca^{2+}]_i$ before and after BzATP 50 μM addition (red arrow) in N-Ca$^{2+}$ and Z-Ca$^{2+}$ in the absence or presence of the P2X7 antagonist A438079 125 nM (aA43); pseudocolors from black to red represent low to high $[Ca2+]_i$, respectively; time frames are indicated in minutes. D) Representative Ca$^{2+}$ fluorescence (Fluo-4) traces from SKOV-3 cells obtained in N-Ca$^{2+}$, Z-Ca$^{2+}$ and Z-Ca$^{2+}$ plus A438079 125nM (aA43). Cells were stimulated with 50 μM BzATP in N-Ca$^{2+}$ or Z-Ca$^{2+}$ extracellular solutions in the presence or absence of the P2X7 receptor antagonist aA43; at the end of the protocol, ionomycin (10μM) and MnCl$_2$ (5mM) were sequentially applied to determine the maximum and minimum levels of intracellular Ca$^{2+}$, respectively. E) Mean ± S.D. of the slope displayed by the sustained component of the Ca$^{2+}$ response induced by BzATP in N-Ca$^{2+}$, Z-Ca2+ and Z-Ca$^{2+}$ plus aA43 observed in SKOV-3 cells. At least 100 cells were analyzed per experiment, n = 3.

proteins as described in methods, the receptor was detected as a broad band of around 43 kDa (Fig 5B), suggesting that the main variant expressed by SKOV-3 cells is not the variant A.

We utilized live-cell Ca$^{2+}$ imaging recordings to obtain functional evidence of P2X7 receptor expression (Fig 5C–5E). In response to 50 μM of BzATP in normal extracellular Ca$^{2+}$ solution (N-Ca$^{2+}$), SKOV-3 cells showed a biphasic response in which the first component was a fast peak followed by a sustained component that did not return to basal levels even after 1.5 min (Fig 5A and 5B). We propose that the initial peak was partially elicited by a P2Y receptor,

probably P2Y11 (Communi D et al, 1991). This receptor is sensitive to BzATP and our group has detected its expression in SKOV-3 cells using RT-PCR. Moreover. we propose that the sustained component depends solely on P2X7 receptor activity. Both zero extracellular $Ca^{2+}$ ($Z$-$Ca^{2+}$) and the presence of the P2X7 receptor antagonist A438079 (aA43-N-$Ca^{2+}$) depleted almost entirely the sustained component of the BzATP-dependent response. To quantitatively compare such described responses, we calculated the slope of the sustained phase in each treatment (Fig 5C). The differences clearly show that $Ca^{2+}$ influx inhibition ($Z$-$Ca^{2+}$) or antagonism of the P2X7 receptor abolished the sustained component of the BzATP-elicited response.

These data prove the presence of P2X7 receptor as well as its functionality in the ovarian carcinoma cell line SKOV-3.

## 5.-P2X7 receptor activity promotes cell migration in OvCar cell lines and maintains a mesenchymal phenotype

To test if P2X7 receptor activity regulates cell migration in metastatic ovarian carcinoma cells, a scratch assay was performed in SKOV-3 cell cultures in the presence of P2X7 receptor ligands. Thus, the incubation with 50 μM of BzATP increased cell migration (141.09 ± 1.82% normalized to control), while the P2X7 antagonists A438079 (125 nM), BBG (100 nM) and oxidized ATP (200 μM) reduced it (50.63 ± 3.04%, 34.95 ± 1.76%, and 45.68 ± 1.32% to control, respectively). To discard the possibility that pharmacological treatments might interfere with cell viability and, therefore, be the reason why cell migration was reduced, we analyzed cell viability through an MTS assay (S2 Fig). We incubated the same number of SKOV-3 cells with either the agonist (BzATP) or antagonists (A438079, BBG and OxATP) for 24 h and measured the absorbance of the tetrazolium salt produced by mitochondrial activity. After normalizing to control, we did not observe any significant differences within treatments. To further confirm that P2X7 activity in SKOV-3 cell migration was mainly mediated by the receptor, we performed the same scratch assay in the CAOV-3 cell line, applying pharmacological agents to analyze P2X7 activity (S3 Fig). Just like we observed with the SKOV-3 cell line, incubation with the P2X7 agonist BzATP significantly increased cell migration in comparison to the control (204.81 ± 14.93%, data normalized to the control). In the same context, A438079, BBG and OxATP significantly decreased cell migration (70.52 ± 4.32%, 67.34 ± 5.90%, and 64.20 ± 7.40% to control, respectively).

To analyze the effect of P2X7 receptor ligands on the SKOV-3 epithelial or mesenchymal phenotype, TEER measurements were recorded daily for 5 consecutive days. Stimulation with 100 μM ATP reduced TEER drastically on day 5 (65.5 ± 0.9 and 31.2 ± 1.2 $\Omega cm^2$ for control and ATP-treated cells, respectively), while the antagonist A438079 substantially increased electrical resistance (118.0 ± 2.4 $\Omega cm^2$) (Fig 6B). From TEER determinations we conclude that P2X7 receptor activity (either basal or induced by external ATP) supports the mesenchymal phenotype displayed by SKOV-3 cells.

The induction of cell migration in ovarian carcinoma cells might be related to EMT. To explore if P2X7 receptor regulates the expression of EMT markers, we analyzed the presence of E-cadherin by western blotting. We incubated SKOV-3 cell cultures for 24 h with the P2X7 agonist BzATP and the antagonists A438079, BBG and OxATP. Pharmacological activation of P2X7 with BzATP abolished the expression of E-cadherin (0.55 ± 0.04, normalized to control) (Fig 6C), whereas the antagonists did not have a significant effect in means of protein detection through western blot (0.82 ± 0.08, 0.97 ± 0.08, and 0.88 ± 0.11 for A438079, BBG and OxATP, respectively). For vimentin, we utilized immunofluorescence. We seeded SKOV-3 cells in cover slides and stimulated with BzATP 50 μM for 24 h and BzATP alone or combined with A438079 for 24 h. BzATP induced an increment in vimentin's signal (481.3 ± 38.27 A.U.).

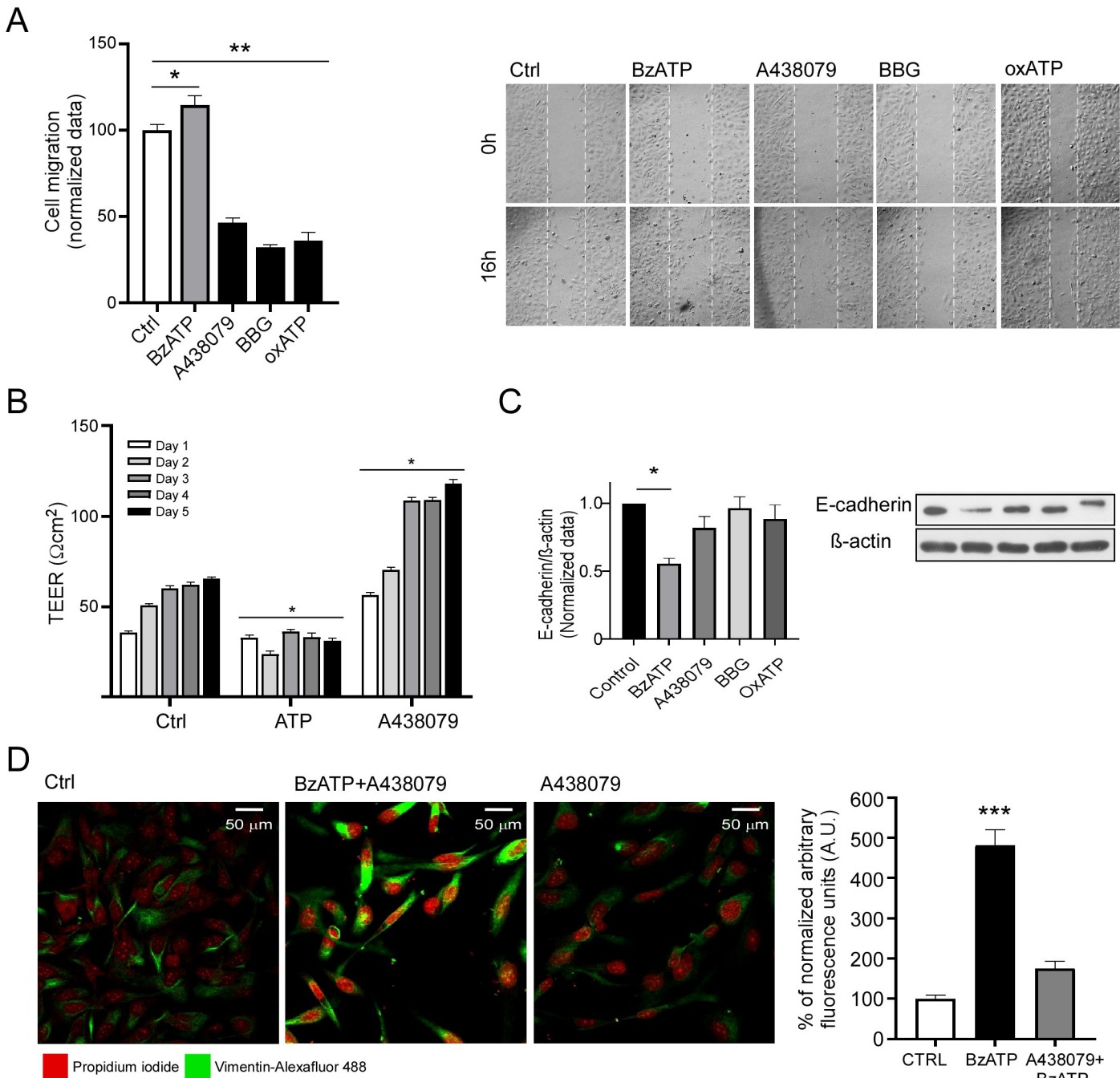

**Fig 6. P2X7 modulates migration and the epithelial-mesenchymal phenotype.** A) Confluent cell cultures of SKOV-3 were stimulated with BzATP 50 μM for 16 h, and migration was measured by the wound healing assay. A series of antagonists were also employed: the specific antagonist A438079 125 nM, non-specific antagonist BBG 200 nM, non-reversible antagonist OxATP 200 μM. B) Transepithelial electrical resistance (TEER) was measured after stimulating with ATP 100 Mm and A438079 125 nM and compared to a control. TEER was measured daily for 5 days, and fresh medium was added daily. C). EMT markers were analyzed by western blot. SKOV-3 at 70% confluence were stimulated with BzATP 50 μM, A438079 125 nM, BBG 200 nM and OxATP 200 μM. In the graphs, bars represent the mean value ± S.E.M. of three different experiments. P < 0.05. D) SKOV-3 cells were treated for 16 h with BzATP 50 μM alone or in combination with A438079 125 nM and compared to a control. Nuclei were stained with propidium iodide, and vimentin was detected with an Alexa Fluor 488 coupled antibody. For the graphs, 50 cells per treatment were analyzed. Bars represent the mean value ± S.E.M. of three different experiments. *p < 0.05.

This effect was abolished when using the antagonist. All of these values were normalized to a control treatment (Fig 6D).

## 6.- Knocking out P2X7 receptor interferes with SKOV-3 cell migration

To have a better certainty of P2X7 receptor role in SKOV-3 cell migration, we generated P2RX7 Knockout SKOV-3 cells (SKOV3-P2X7$^{KO}$) using a CRISP-CAS9 based system (Santa Cruz Biotech, SC-400780-NIC). First, we characterized our experimental system by RT-PCR, employing primers previously validated and using the parental SKOV-3 cell line as a positive control (Fig 7A). We employed Ca$^{2+}$ imaging recordings as a direct approach to identify changes in the receptor's function, as we showed previously. Stimulation with BzATP on the SKOV3-P2X7$^{KO}$ cells did not induce the sustained component elicited in the parental cell line system (Fig 7B). To verify the participation of P2X7 in cell migration, we performed wound-healing assay. Once again, we used the parental SKOV-3 cell line as a positive control and we observed a prominent increase in cell migration (197.83 ± 10.26%, normalized to control) (Fig 7C), while this effect was not observed when we employed the SKOV3-P2X7$^{KO}$ cells (111.90 ± 5.40%, normalized to its control).

## 7.-P2X7 promotes tumor growth in a zebrafish xenograft model

To probe whether P2X7 receptor has a role in SKOV-3 cells spread in an *in vivo* model, SKOV-3 cells were transfected with a lentiviral plasmid (pKLO1) carrying the coding sequence for yellow fluorescent protein (YFP). Positive cells were selected by a puromycin resistance cassette present in the vector, which exhibited a stable expression of GFP. These cells were named SKOV-3$^{GFP}$. SKOV-3$^{GFP}$ cells were incubated for 2 h with either the P2X7 non-reversible antagonist OxATP (200 μM) or media alone, and 200 cells were microinjected into a 16 h-old zebrafish embryo. Photographs were taken immediately after injection (time 0) and 48 h after, and the cell cumulus was measured (drawing a straight line from one side of the tumor to the other, 3 times per embryo). Incubating the SKOV-3$^{GFP}$ cells with OxATP impeded the growth and/or displacement of the injected tumor cells (-1.23 ± 7.09%, comparing hour 48 to time 0), while the non-treated cells had significant growth and/or displacement (20.83 ± 5.84%) (Fig 8).

## Discussion

Research has proven that the extracellular milieu of cancerous tissue contains high concentrations of ATP [6], which is an important mediator for the malignant phenotype of many cancers [47]. In healthy models, ATP has also been shown to modulate the migration of CD$^{34+}$ stem cells [48], microglia [49], mesenchymal stem cells [50] and corneal epithelial cells [51], among others. Given the high concentrations of ATP in the tumor microenvironment and that cancer cells are highly capable of metastasizing, it is no surprise that cancer cells use ATP as a mobilization signal, as has been reported in different cancer models, such as lung [52], cervix and breast [53]. Previously, our group has demonstrated that the ovarian carcinoma cell line SKOV-3 continuously releases ATP to the extracellular milieu [23] and exerts a basal activation of migration-related pathways, like AKT. In this work, we proved through gene expression patterns, cytoskeletal arrangements, phenotypical modifications and cell physiology that ATP depletion has a negative effect on cell migration. Purinergic receptors have been detected in most cancers and their importance in malignity has been highlighted [54]. Therefore, it is logical to think that a P2 receptor mediating ATP promotes migration.

P2X7 is a promising drug target in biomedical research that has gained attention since its first descriptions as a cytolytic promoter [30]. Research has helped to understand the functions

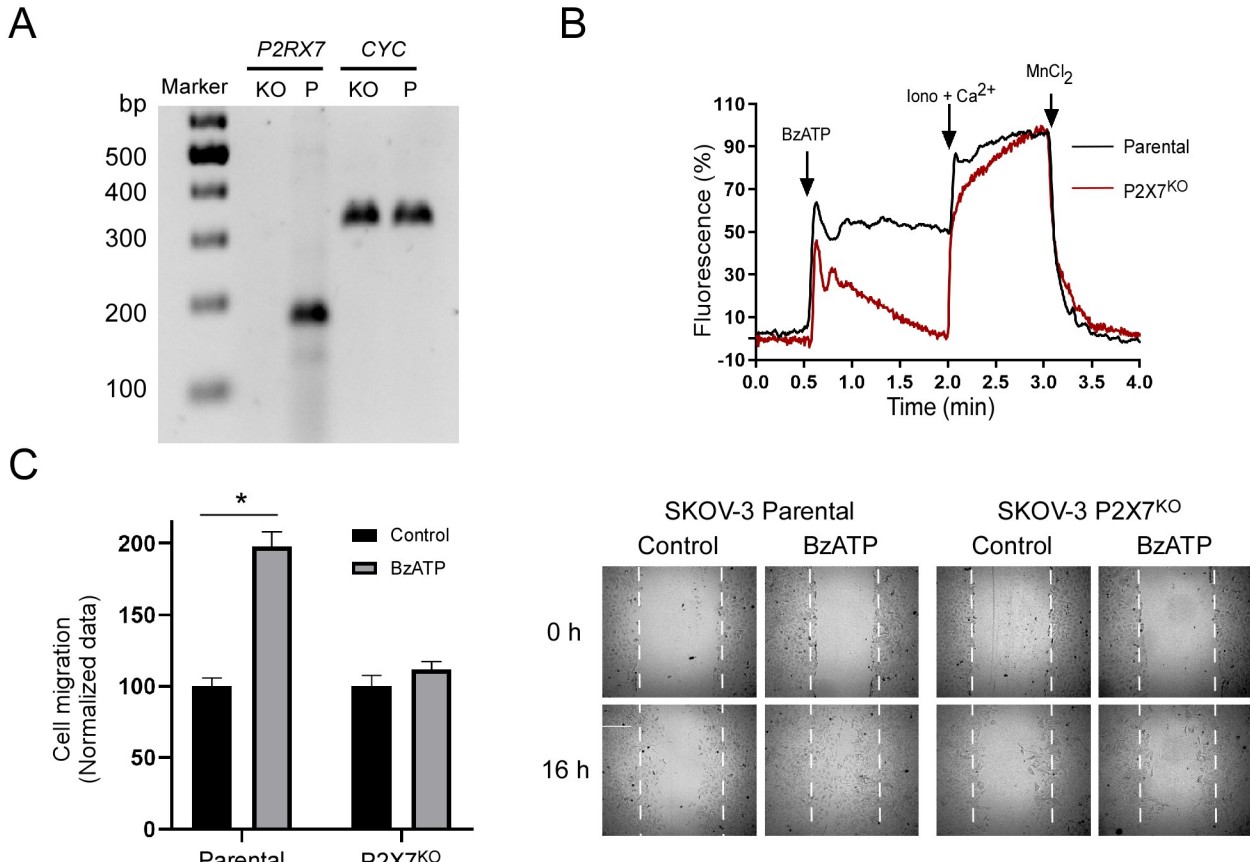

**Fig 7. P2X7 knock-out abolishes cell migration.** A) *P2RX7* and constitutive (*CYC*) transcripts were analyzed by RT-PCR in the SKOV-3 parental (Parental) and SKOV3-P2X7$^{KO}$ (P2X7$^{KO}$) cells, amplicons were analyzed in agarose 1.5% gels, primers were previously validated. B) Representative Ca$^{2+}$ fluorescence (Fluo-4) traces from either SKOV-Parental cell line and SKOV3-P2X7$^{KO}$ cells. Cells were stimulated with 50μM BzATP in N-Ca$^{2+}$ extracellular solution; at the end of the protocol, ionomycin (10μM) and MnCl$_2$ (5mM) were sequentially applied to determine the maximum and minimum levels of intracellular Ca$^{2+}$, respectively. C) SKOV-Parental line and SKOV3-P2X7$^{KO}$ confluent cultures were stimulated with BzATP 50μM during 16h and cell migration was measured by the scratch assay. Representative pictures are shown at time 0h and 16h. In the graphs, bars represent the mean value ± S.E.M. of three different experiments. *p < 0.05.

and relevance of P2X7 in cancer. For instance, it has been reported that P2X7 plays a role in the maintenance of a malignant phenotype of different cancers, such as prostate [55], lung [56], osteosarcoma [57], pancreas [58], breast [59] and ovarian carcinoma [23]. This is consistent with the results of our qPCR analysis, which revealed a substantial increase in the *P2RX7* transcript when comparing cell lines derived from transformed tissues (in this case, SKOV-3) and the non-cancerous HOSE cell line. We examined sequencing datasets and saw a significant increment in P2RX7 in metastatic tissues compared to primary tumors, indicating a possible role in ovarian carcinoma cell migration. Many papers support this idea, given that P2X7 has been associated with migration and metastasis promotion in other cancer types, such as pancreas [27], colon [29], and breast cancer [60].

To provide a clear overview of our system, we characterized *P2RX7* expression, the functional role of P2X7 receptors depends on the splicing variant expressed. Homomeric channels formed with P2X7A subunits are related with cytotoxic effects of receptor activation [61] while isoforms lacking COOH-end tail, loss the cytotoxic activity. Between the most outstanding variants P2X7B [31] and P2X7J [17] are two of the best characterized. P2X7B is a truncated

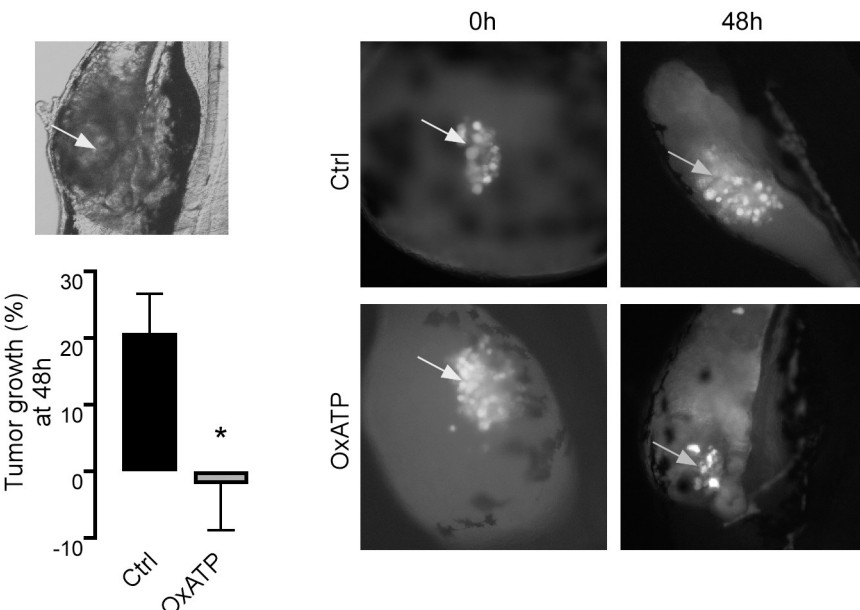

**Fig 8. Pharmacological antagonism of P2X7 impedes tumor growth in vivo.** SKOV-3 cells were incubated with OxATP 200 μM for 2 hours prior to being injected into 16 hour-old zebrafish embryos. Pictures were taken at 0 h and 48 h. Tumor size was measured three times per zebrafish, and 4 animals per treatment were used. In the graph, bars represent the mean value ± S.E.M. *p < 0.05.

isoform lacking COOH-end, whose expression results in functional channels but without the ability to induce apoptotic cell death, this variant can support cell proliferation and migration by self or regulating P2X7A function by forming heteromeric channels [31]. On the other hand P2X7J is a short variant lacking the second transmembrane spam that function as dominant negative resulting in loss of cell death-induction and favoring cell survivance [17].

We did PCR with primers designed to recognize the region of the transcript corresponding to the extracellular loop of the P2X7 receptor (aa 61–67), a fundamental site for the function of the P2X7 receptor, as it forms part of the ATP binding site (Lys$^{64}$ and Lys$^{66}$) [62]; and is a common region for almost all the splicing variants of *P2RX7* transcript [31]. Ovarian carcinoma cell lines SKOV-3 (Fig 5A), CAOV-3 and SW626 resulted positive to *P2RX7* transcript expression (S3 Fig). Moreover, we determined at protein level that metastatic SKOV-3 cells are expressing two variants of P2X7 receptor, the variant A (~75 kDa) and a variant of ~43 kDa lacking COOH-end, probably variant B [31]. According with our western blot observations the short variant is the predominant (Fig 5B). This result suggested that all the actions mediated by heteromeric P2X7 receptors.

P2X7 is a channel that differs from that of other P2X receptors because it does not desensitize when the agonist is bound [11]; thus, we implemented a pharmacological characterization using a specific agonist (BzATP) and recorded calcium mobilization and plotted calcium entry as a function of time, observing a sustained signal characteristic of P2X7 activity (Fig 5C and 5D).

EMT is a dynamic trans-differentiation process that is highly active during embryogenesis, tissue repair and metastasis. Two EMT markers, E-cadherin and vimentin, have been reported to decrease significantly and increase, respectively, allowing cells to gain motility while reducing intercellular adhesion [63]. Our results point toward EMT activation, as we measured

changes in E-cadherin and vimentin, as well as a less epithelial/more migratory phenotype (. Although the role of P2X7 in EMT activation has not been formally characterized, many reports have found markers of an EMT-like process upon P2X7 activation [64]. There are also several reports concerning the importance of extracellular purines in EMT [65]. However, no purinergic element has emerged as a major player, suggesting the possibility that purinergic receptors work in a cooperative manner. Taking this into consideration, we can suggest that some purinergic receptors are orchestrated in OvCar to promote cell migration and trans-differentiation, given that we have previously reported the role of P2Y2 in promoting cell migration [66] and how A2B impedes such process [67]. A thorough understanding of purine function in EMT regulation is evidently necessary, as this process has been linked to sustaining a malignant phenotype, promoting chemo-resistance and cancer recurrence [68].

In the present study we aimed to characterize the autocrine effects of the axis: extracellular ATP/P2X7 receptor on the cell phenotype; but purinergic mechanisms in cancer involves also the interactions of tumor cells with the microenvironment and in general with the host cells; in this case in vivo approaches takes relevance, for example the systemic administration of P2X7 receptor antagonists A438079 and AZ10606120 in an model of pancreatic cancer incremented the spread of carcinoma cells and reduced the survival rates [69]. Furthermore, in a model of colitis-associated cancer, pharmacological or genetic inhibition of P2X7 receptor revealed that P2X7 receptor restrain proinflammatory responses but also is an early tumor suppressor [70]. These evidence shows the role of P2X7 receptor involves also the specific identity of the variants of the receptor expressed in a tissue, the dynamic of the ligand that is strongly related with ectonucleotidasas expression but also the interactions with immune cells; some integrative reviews reflect on this topic [7–10], although the present study is aimed on autocrine-paracrine actions of ATP-P2X7 axis, integrative data should be considered to the potential target of P2X7 receptor as therapeutic target.

Altogether, our results consistently confirm that extracellular ATP is an important factor that promotes a malignant phenotype in an autocrine-paracrine way and that P2X7 is a key modulator of its effects in ovarian carcinoma.

## Supporting information

**S1 Fig. ATP abolishment decreases cell migration.**
(PDF)

**S2 Fig. P2X7 pharmacological stimulation does not affect cell viability.**
(PDF)

**S3 Fig. P2X7 is functionally expressed in ovarian carcinoma cell lines.**
(PDF)

**S1 Table. Differentially expressed purinergic receptors in patients-derived samples.**
(DOCX)

**S2 Table. Ovarian cancer samples tested for P2X7 immunostaining in human protein atlas (https://www.proteinatlas.org/ENSG00000089041-P2RX7/pathology/ovarian+cancer).**
(DOCX)

**S1 Raw images.**
(PDF)

## Acknowledgments

José David Núñez Ríos is a doctoral student from the Programa de Doctorado en Ciencias Biomédicas, Universidad Nacional Autónoma de México (UNAM) and has received CONAH-CYT fellowship CVU #1044618. We are grateful to Jessica Gonzalez Norris for proofreading this manuscript. We are also grateful to Ing. Nydia Hernández Ríos, Biol Maria Eugenia Ramos Aguilar, Dr Olivia Vázquez-Martínez and Nuri Aranda López for technical assistance.

## Author Contributions

**Conceptualization:** José David Nuñez-Ríos, Francisco G. Vázquez-Cuevas.

**Data curation:** José David Nuñez-Ríos, Mauricio Reyna-Jeldes, Esperanza Mata-Martínez, Anaí del Rocío Campos-Contreras, Iván Lazcano-Sánchez, Adriana González-Gallardo, Claudio Coddou, Francisco G. Vázquez-Cuevas.

**Formal analysis:** José David Nuñez-Ríos, Esperanza Mata-Martínez, Adriana González-Gallardo, Mauricio Díaz-Muñoz, Claudio Coddou, Francisco G. Vázquez-Cuevas.

**Funding acquisition:** Claudio Coddou, Francisco G. Vázquez-Cuevas.

**Investigation:** José David Nuñez-Ríos, Mauricio Reyna-Jeldes, Esperanza Mata-Martínez, Anaí del Rocío Campos-Contreras, Iván Lazcano-Sánchez, Adriana González-Gallardo, Mauricio Díaz-Muñoz, Claudio Coddou, Francisco G. Vázquez-Cuevas.

**Methodology:** José David Nuñez-Ríos, Mauricio Reyna-Jeldes, Esperanza Mata-Martínez, Anaí del Rocío Campos-Contreras, Iván Lazcano-Sánchez, Adriana González-Gallardo, Francisco G. Vázquez-Cuevas.

**Project administration:** Francisco G. Vázquez-Cuevas.

**Supervision:** Mauricio Díaz-Muñoz, Claudio Coddou, Francisco G. Vázquez-Cuevas.

**Validation:** Mauricio Reyna-Jeldes.

**Visualization:** Francisco G. Vázquez-Cuevas.

**Writing – original draft:** José David Nuñez-Ríos, Francisco G. Vázquez-Cuevas.

**Writing – review & editing:** José David Nuñez-Ríos, Anaí del Rocío Campos-Contreras, Claudio Coddou, Francisco G. Vázquez-Cuevas.

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
