## [Decision Letter · Decision Letter 0]

4 Mar 2024

PONE-D-24-01608Extracellular ATP/P2X7 receptor, a regulatory axis of migration in ovarian carcinoma-derived cells.PLOS ONE

Dear Dr. Vázquez-Cuevas,

Thank you for submitting your manuscript to PLOS ONE. After careful consideration, we feel that it has merit but does not fully meet PLOS ONE’s publication criteria as it currently stands. Therefore, we invite you to submit a revised version of the manuscript that addresses the points raised during the review process.

We look forward to receiving your revised manuscript.

Kind regards,

Altaf Mohammed

Academic Editor

PLOS ONE

“José David Núñez Ríos is a doctoral student from the Programa de Doctorado en Ciencias Biomédicas, Universidad Nacional Autónoma de México (UNAM) and has received CONAHCYT fellowship CVU #1044618. We are grateful to Jessica Gonzalez Norris for proofreading this manuscript. We are also grateful to Ing. Nydia Hernández Ríos, Biol Maria Eugenia Ramos Aguilar and Nuri Aranda López for technical assistance. This study was supported by the grant IN205223 from PAPIIT-UNAM; FONDECYT Regular Grant 1161490, FONDEQUIP EQM140100, intramural grant “Núcleo para el estudio del cáncer a nivel básico, aplicado y clínico”, VRIDT, UCN and Millennium Nucleus for the Study of Pain. MiNuSPain is a Millennium Nucleus supported by the Millennium Science Initiative of the Ministry of Science, Technology, Knowledge and Innovation (Chile). Funders had no role in study design, data collection and analysis, decision to publish, or preparation of this article.”

“This study was supported by the grant IN205223 from PAPIIT-UNAM; FONDECYT Regular Grant 1161490, FONDEQUIP EQM140100, intramural grant “Núcleo para el estudio del cáncer a nivel básico, aplicado y clínico”, VRIDT, UCN and Millennium Nucleus for the Study of Pain. MiNuSPain is a Millennium Nucleus supported by the Millennium Science Initiative of the Ministry of Science, Technology, Knowledge and Innovation (Chile). Funders had no role in study design, data collection and analysis, decision to publish, or preparation of this article.”

4. We notice that your supplementary figures are uploaded with the file type 'Figure'. Please amend the file type to 'Supporting Information'. Please ensure that each Supporting Information file has a legend listed in the manuscript after the references list.

Reviewers' comments:

Reviewer's Responses to Questions

**Comments to the Author**

1. Is the manuscript technically sound, and do the data support the conclusions?

Reviewer #1: Yes

Reviewer #2: Yes

2. Has the statistical analysis been performed appropriately and rigorously? 

Reviewer #1: I Don't Know

Reviewer #2: Yes

3. Have the authors made all data underlying the findings in their manuscript fully available?

Reviewer #1: Yes

Reviewer #2: Yes

4. Is the manuscript presented in an intelligible fashion and written in standard English?

Reviewer #1: Yes

Reviewer #2: Yes

5. Review Comments to the Author

Reviewer #1: Authors elucidated pro-metastatic role of P2X7R in ovarian carcinoma cell line. Authors observed a decrease in cell migration and an increase in transepithelial electrical resistance and cell markers, suggesting P2X7R expression maintains a mesenchymal phenotype. Transcript levels analysis of some P2 receptors lead to three-fold higher P2RX7 levels in SKOV-3 cells than in a healthy cell line, namely HOSE6-3. Through bioinformatic analysis, authors identified a higher expression of the P2RX7 transcript in metastatic tissues than in primary tumors; thus, P2X7 seems to be a promising effector for the malignant phenotype. Subsequently, authors were demonstrated that presence and functionality of the P2X7 receptor in SKOV-3 cells and showed through pharmacological approaches, CRISPR-based knock-out system and in vivo Zebrafish embryonic assays role of P2X7 receptor is a regulator for cancer cell migration/invasion and thus a potential drug target. Overall, authors conclusions were drawn by experimental results on a metastatic ovarian cell line.

Comments:

1. Previously, many in vitro studies with different organ-site cancer cell lines shown the role of P2X7R in cell migration and invasion (eg. Molecular Cancer volume 14, Article number: 203, 2015; Mol Biol Cell. 2005;16:3260–72; FASEB J. 2010;24:3393–404). In several of published studies suggest P2X7R isoform A involvement in migration. Does authors tested whether Ovarian cancer SKOV-3 P2X7R over-expressed transcript is specific Isoform? Discuss in the text.

2. Figure 3C protein atlas data – Figure show very small deviation for survival (PFS) with Low: 18.83 months vs High: 18.65 months, just few days difference with p=0.044. Does authors analyze overall survival (OS) of patients with different stages with low and high P2X7R expressions. Please include these data and discuss significance PFS vs OS.

3. Although, authors Figure 8 experiments with Zebrafish interesting, how authors selected oxATP or P2X7R antagonist concentrations? It should have better used two or three doses!!

4. Even though, in vitro and some in vivo studies shown promoting effect of P2X7R over-expression; whereas in well-established experimental animal models either pharmacological targeting (Oncotarget. 2017 Nov 17; 8(58): 97822–97834) and genetic ablation (Cancer Res (2015) 75 (5): 835–845) of P2X7R lead to enhanced tumor growth, thus clearly suggesting pro-tumorigenic role. Authors should cite these articles and discuss in the discussion of part of text.

Reviewer #2: PONE-D-24-01608

Manuscript Title: Title: Extracellular ATP/P2X7 receptor, a regulatory axis of migration in ovarian carcinoma derived cells.

General comments: The authors of this study reported on the role of the P2X7 receptor in metastasis of ovarian cancer. Although this receptor's involvement in metastasis has been proven in other cancers, such as breast, pancreas, lung and melanoma, the authors suggest that it also has the same role in ovarian cancer. To prove their hypothesis, the authors used a single ovarian cancer cell line and studied a few human tissue samples. However, to provide more clarity on the role of P2X7 in ovarian cancer, they need to provide additional information, such as testing other types of ovarian cancer cell lines (PA-1, SW626, Caov-3), studying more human samples (a human samples panel), and performing additional experiments.

Specific questions:

It is unclear which specific gene mutations the tested ovarian cancer cell line has, as the authors did not mention it. Furthermore, the authors need to compare the data obtained from this cell line with other primary tumors or adenocarcinomas such as SW626 and Caov-3. They also need to provide data on the tissue samples collected from the metastatic site.

The authors need to provide details on whether the biopsies tested for P2X7 receptor expression were collected from the primary tumor site or the metastatic site. Additionally, the authors need to perform experiments with other primary tumor-derived ovarian cancer cell lines to show the difference in the data obtained using more than one cell line. The authors also need to address whether the P2X7 receptor is responsible for the aggressive nature of the serous histology type of ovarian cancer or if the serous type at the adenocarcinoma stage also expresses high levels of this receptor. In either case, they need to include experimental evidence for their explanation.

They also need to provide experimental evidence to show if this receptor expression causes the serous type to metastasize.

The authors need to explain how they optimized the doses of P2X7 antagonists A438079 125 nM (aA43) and non-reversible antagonist OxATP (200 μM).

The presence of the P2X7 receptor on immune cells such as neurons, DCs, macrophages, and B and T cells, etc., could influence the metastatic behavior of the tumor cells. Therefore, the authors need to discuss how the tumor cells interact with other cells that express this receptor and provide any data from human tissue samples.

Additionally, preclinical immune-competent animal experiments are required to understand how this receptor interacts with tumor cells in causing metastasis.

Finally, the authors need to provide data on the expression of P2X7 from lesions, adenomas, adenocarcinoma to metastatic tissues. This will give a clear indication of when this receptor is expressed during which stage of cancer development. They also need to explain the clinical significance of their observations reported in this paper.

6. PLOS authors have the option to publish the peer review history of their article (what does this mean?). If published, this will include your full peer review and any attached files.

Reviewer #1: **Yes: **Chinthalapally V. Rao

Reviewer #2: No

---

## [Author Response · Author response to Decision Letter 0]

16 Apr 2024

Response to reviewers

Reviewer #1: Authors elucidated pro-metastatic role of P2X7R in ovarian carcinoma cell line. Authors observed a decrease in cell migration and an increase in transepithelial electrical resistance and cell markers, suggesting P2X7R expression maintains a mesenchymal phenotype. Transcript levels analysis of some P2 receptors lead to three-fold higher P2RX7 levels in SKOV-3 cells than in a healthy cell line, namely HOSE6-3. Through bioinformatic analysis, authors identified a higher expression of the P2RX7 transcript in metastatic tissues than in primary tumors; thus, P2X7 seems to be a promising effector for the malignant phenotype. Subsequently, authors were demonstrated that presence and functionality of the P2X7 receptor in SKOV-3 cells and showed through pharmacological approaches, CRISPR-based knock-out system and in vivo Zebrafish embryonic assays role of P2X7 receptor is a regulator for cancer cell migration/invasion and thus a potential drug target. Overall, authors conclusions were drawn by experimental results on a metastatic ovarian cell line.

Comments:

1. Previously, many in vitro studies with different organ-site cancer cell lines shown the role of P2X7R in cell migration and invasion (eg. Molecular Cancer volume 14, Article number: 203, 2015; Mol Biol Cell. 2005;16:3260–72; FASEB J. 2010;24:3393–404). In several of published studies suggest P2X7R isoform A involvement in migration. Does authors tested whether Ovarian cancer SKOV-3 P2X7R over-expressed transcript is specific Isoform? Discuss in the text.

Answer:

P2X7 receptor has a complex posttranscriptional processing resulting in various isoforms. Some of these have been functionally characterized. Homomeric channels formed with P2X7A subunits are related with cytotoxic effects of receptor activation (DOI: 10.1152/ajpcell.00256.2004). P2X7A has a long COOH-end that enables the interaction with other membrane proteins to induce the assembly of the “megapore”, a large conductance entity that allows the passage of molecules as large as 1 kDa. The activation of this megapore has been related with apoptotic cell death as result of Ca2+ overload and activation of the intrinsic apoptosome pathway (DOI: 10.1074/jbc.272.9.5482; DOI: 10.1152/ajpcell.00256.2004). Isoforms lacking COOH-end tail, loss the cytotoxic activity. Between the most outstanding variants P2X7B (DOI: 10.1096/fj.09-153601 ) and P2X7J (DOI: 10.1074/jbc.M602999200 ) are two of the best characterized. P2X7B is a truncated isoform lacking COOH-end, whose expression results in functional channels but without the ability to induce apoptotic cell death; this variant can support cell proliferation and migration by itself, regulating P2X7A function by forming heteromeric channels (DOI: 10.1096/fj.09-153601). On the other hand, P2X7J is a short variant lacking the second transmembrane spam that function as dominant negative resulting in loss of cell death-induction and favoring cell survivance (DOI: 10.1074/jbc.M602999200). 

In the present study, to detect the expression of all the P2RX7 transcripts in the ovarian carcinoma cell line SKOV-3, a pair of primers directed against the conserved 5´region of the transcript were designed. For that, an alignment of variants A (Accession number Y09561.1), B (AY847298.1), C (AY847299.1), D (AY847300.1), E (AY847301.1), H (AY847304.1), F (AY847302.1) and J (DQ399293.1) was made. These primers are located in the nucleotide number 277-291 (forward) and 471-452 (reverse) with respect to the variant A sequence; the oligonucleotide forward corresponds to a portion of exon 2, and the oligo reverse a portion between the exons 3 and 4 of the P2RX7 locus (ENSG00000089041); thus, the amplicon obtained is produced by transcript expression, but it is no possible discriminate which variant is expressed.

To attend the reviewer question, in an attempt to elucidate which variant or variants of the receptor are expressed by SKOV-3, proteins located in the plasmatic membrane were purified by biotinylation with a non-permeable reagent (Thermo Scientific), followed by affinity isolation with Sepharose-Avidin beads (Cell Signaling Technology); then, P2X7 receptor was identified by western blotting using an antibody directed against the Extracellular Loop of the receptor (Alomone APR-008) or to COOH-end (Alomone APR-004).

In the western blot analysis of membranal proteins from SKOV-3 cells, COOH-end antibody revealed a faint band of 75 kDa, corresponding with the isoform A while extracellular-loop-P2X7 antibody revealed a very strong signal corresponding to around 43 kDa. These results suggested that SKOV-3 cells express at least two P2X7 variants, variant A (75 Da) and a variant of around 43 kDa, probably variant B, whose expression level is predominant. These results were included in the Figure 5B of the new version of the manuscript and described in lines 437-442 of the Results section and lines 594-603 and 606-612 in the Discussion. Also, the methods employed were also described in the appropriate section of the manuscript, lines 226-236.

Therefore, the new evidence suggests that Skov-3 express at least two P2X7 variants, variant A (75 Da) and a variant of around 43 kDa lacking the COOH-end, probably variant B, whose expression level is predominant. 

2. Figure 3C protein atlas data – Figure show very small deviation for survival (PFS) with Low: 18.83 months vs High: 18.65 months, just few days difference with p=0.044. Does authors analyze overall survival (OS) of patients with different stages with low and high P2X7R expressions. Please include these data and discuss significance PFS vs OS.

Answer: 

Thank you for pointing this issue. While editing the pictures coming from the Kaplan-Meier plotter website, we made a untimely mistake and wrote 18.65 months instead of 16.65 months for PFS in the 2+3 stages of Serous Ovarian carcinoma. Below you can find a permanent link proving such information. 

https://kmplot.com/analysis/index.php?p=view&pa_id=21284063&show=bZFNboQwDIXvwrqqNItuehnLBGewCHFkO8y0o969gUULiO33np__Xp3hQoAGzjF2n13EZNS9dTbKA2YaGHOjrvUfcp7xeYI9kT1Q5-sEkAhWdeEF07kQw9QKB1CKpJQDnSNKEofEmcBHDlMms2a5HWSnp0MvabhWjL_X2Nv7x0EL1VxmmJFzW9_TdevShq4ERYydZbuGFOX76McJ6U552Nt68RafKK7GolLCSGHa9fhjgCk98Mt2WjWCBZWxTwRRFIKMor63BFEFK9TOvj3pwFdsks8bheqy_Xn7wc8v

We are analyzing the role of P2X7 in promoting cancer cell migration in vitro, which could be extrapolated to cancer progression and metastases. In the PFS plots, we saw that the higher the receptor expression the lower the PFS value, and keeping in mind that the PFS is the time that a patient lives with the disease but does not get worse, we could infer that higher P2RX7 transcript levels correspond with a more likely disease progression and metastatic events. On behalf of the role of P2RX7 transcript levels on OS, we obtained the next graphs: 

Here we can observe that the higher the P2RX7 expression, the higher the OS, making it a predictor for better prognosis. At a first glance, it might seem contradictory that PFS and OS for high P2RX7 levels have such different values, but a review of clinical studies (https://doi.org/10.7150/jca.32205) showed that these values could be determined due to different therapeutic mechanisms. As we mentioned previously in this text, there are different splicing variants and such variants have different effects on ongoing cell biology of this process. Given the limitations on the Kaplan Meier Plotter platform, it is impossible to discriminate weather it was one splicing variant or the other along the duration of the studies. 

3. Although, authors Figure 8 experiments with Zebrafish interesting, how authors selected oxATP or P2X7R antagonist concentrations? It should have better used two or three doses!!

Answer: 

We decided to use oxATP 200 uM in a single dose given that it is accepted as a well characterized irreversible P2X7 antagonist that form covalent bond with the receptor. Its mechanism of action was described in the early 90s and which pIC50 was originally published by Di Virgilio’s group (Murgia M., et al 1993). 

4. Even though, in vitro and some in vivo studies shown promoting effect of P2X7R over-expression; whereas in well-established experimental animal models either pharmacological targeting (Oncotarget. 2017 Nov 17; 8(58): 97822–97834) and genetic ablation (Cancer Res (2015) 75 (5): 835–845) of P2X7R lead to enhanced tumor growth, thus clearly suggesting pro-tumorigenic role. Authors should cite these articles and discuss in the discussion of part of text.

The mentioned articles contain important contributions for understanding P2X7 role in cancer and were included in the discussion section, lines 629-639.

Reviewer #2: PONE-D-24-01608

Manuscript Title: Title: Extracellular ATP/P2X7 receptor, a regulatory axis of migration in ovarian carcinoma derived cells.

General comments: The authors of this study reported on the role of the P2X7 receptor in metastasis of ovarian cancer. Although this receptor's involvement in metastasis has been proven in other cancers, such as breast, pancreas, lung and melanoma, the authors suggest that it also has the same role in ovarian cancer. To prove their hypothesis, the authors used a single ovarian cancer cell line and studied a few human tissue samples. However, to provide more clarity on the role of P2X7 in ovarian cancer, they need to provide additional information, such as testing other types of ovarian cancer cell lines (PA-1, SW626, Caov-3), studying more human samples (a human samples panel), and performing additional experiments.

Specific questions:

1.- It is unclear which specific gene mutations the tested ovarian cancer cell line has, as the authors did not mention it. 

Answer: 

Thank you so much by your comments that clearly improve our manuscript. 

SKOV-3 (ATCC ID HTB-77) cell line is a model of metastatic serous ovarian carcinoma, it was obtained from a white 64 years old woman ascitic material, being hypodiploid with 43 chromosomes in approximately 63% of the cells. This line carries mutations in TP53 and in the subunit alpha of PIK3CA; a detailed description of mutations in genes relevant in cancer is available (DOI: 10.1158/1535-7163.MCT-06-0433). CAOV-3 (ATCC HTB-75) cells were isolated from a primary ovarian adenocarcinoma of a white 54-year-old patient. This information was added in Materials and Methods section, lines 127 to 132.

2.- The authors need to compare the data obtained from this cell line with other primary tumors or adenocarcinomas such as SW626 and Caov-3. 

Indeed, the expression of transcript of P2RX7 receptor, Ca2+ mobilization end cell migration induction mediated by P2X7 agonist were also analyzed in CAOV-3 and SW-626 cell lines. These results were included in the manuscript and placed as a figure in the supplementary material (lines 331 to 336; 487-493 and 607-608; supplementary material S4). 

Although the three lines are from ovarian adenocarcinoma, CAOV-3 and SW-626 were isolated from primary tumors while SKOV-3 was isolated from ascites, in consequence, the last one present mesenchymal characteristic such as spontaneous migration ability. This fact made that the three cell lines are not strictly comparable; rather, they are modeling different stages of the disease. The effect on cell migration was much clearer and more conspicuous in SKOV-3 cells; this fact was of special interest because cells isolated from ascites are of particular clinical interest by their spreading potential (DOI: 10.3389/fendo.2022.886533; DOI: 10.3390/ijms23116215). This is the reason why most of the study was developed in this model. In fact, one of main conclusions of this study is that the axis ATP/P2X7 contributes with the maintenance of mesenchymal characteristics in this type of cells.

3.-The authors need to provide details on whether the biopsies tested for P2X7 receptor expression were collected from the primary tumor site or the metastatic site. They also need to provide data on the tissue samples collected from the metastatic site.

All the data about biopsies were obtained from the Human Protein Atlas. According with this public site all the samples correspond to tumoral biopsies; all the available data about this samples were placed as supplementary material (S5 Table).

In the same sense of the reviewer´s observation, we analyzed comparatively cells from the original tumor with cells from metastatic tissues. For that, our analysis employing public databases consisted in exploring GEO by using the validated GEO2R tool. Within such platform, we found three datasets that compare the differential expression of P2RX7 levels in the metastatic site in comparison to the primary tumor. From the GSE73091 set, we compared ascitic fluid samples to primary tumor samples coming from patients with High grade serous ovarian carcinoma; in a similar context the GSE222982, contains data coming from patients with HGSOC as well; while the GSE73091 come from patients with a Low grade serous ovarian carcinoma, with a similar behavior to the High grade type, thus the higher expression of P2RX7 on metastatic tissues are a characteristic of the OvCar disease instead of just a marker of the high grade histology. This information is added into the article (lines 410 to 420 and Table S3).

4.-Additionally, the authors need to perform experiments with other primary tumor-derived ovarian cancer cell lines to show the difference in the data obtained using more than one cell line.

Answer (please, see the answer of commentary 2)

5.-The authors also need to address whether the P2X7 receptor is responsible for the aggressive nature of the serous histology type of ovarian cancer or if the serous type at the adenocarcinoma stage also expresses high levels of this receptor. In either case, they need to include experimental evidence for their explanation. They also need to provide experimental evidence to show if this receptor expression causes the serous type to metastasize.

Ovarian serous carcinoma is the most aggressive carcinoma, to analyze if a correlation exists between serous carcinoma progression and the presence of P2X7 receptor, we use public database. Our analysis from Human Protein Atlas showed that the serous tumors analyzed are positive to P2X7 expression (Lines 385-387 and supplementary material S5). 

On the other hand, Kaplan Meier Database showed that the probability of survival for patients with tumor Serous (in general), Serous stage 2+3 and Serous stage 3+4) is lesser for patients with high expression level of P2RX7 transcript than those with low expression level (see Figure 3 of our manuscript).

Moreover, by using GEO database we analyzed comparatively cells from the original tumor with cells from metastatic tissue. For that, our analysis employing public databases consisted in exploring GEO and using the validated GEO2R tool. Within such platform, we found three datasets that compare the differential expression of P2RX7 levels in the metastatic site in comparison to the primary tumor. From the GSE73091 set, we compared ascitic fluid samples to primary tumor samples coming from patients with High Grade Serous Ovarian Carcinoma (HGSOC); in a similar context the GSE222982, contains data coming from patients with HGSOC as well; while the GSE73091 come from patients with a low grade serous ovarian carcinoma, with a similar behavior to the high grade type, thus the higher expression of P2RX7 on metastatic tissues are a characteristic of the OvCar disease instead of just a marker of the high grade histology. This information was incorporated into the article (lines 408 to 418 and Table S3).

Taking together these data suggest that P2X7 receptor expression is relevant for serous carcinoma progression and justify the in vitro characterization of the cellular implications of its function, mainly on cell migration ability.

6.-The authors need to explain how they optimized the doses of P2X7 antagonists A438079 125 nM (aA43) and non-reversible antagonist OxATP (200 μM).

Answer: 

We decided to use Oxidized-ATP 200 uM in a single dose given that it is a well characterized irreversible P2X7 receptor antagonist that form covalent bond with the receptor. Its inhibitory mechanism was described in the early 90s

---

## [Decision Letter · Decision Letter 1]

7 May 2024

Extracellular ATP/P2X7 receptor, a regulatory axis of migration in ovarian carcinoma-derived cells.

PONE-D-24-01608R1

Dear Dr. Vázquez-Cuevas,

We’re pleased to inform you that your manuscript has been judged scientifically suitable for publication and will be formally accepted for publication once it meets all outstanding technical requirements.

Kind regards,

Altaf Mohammed

Academic Editor

PLOS ONE

**Comments to the Author**

1. If the authors have adequately addressed your comments raised in a previous round of review and you feel that this manuscript is now acceptable for publication, you may indicate that here to bypass the “Comments to the Author” section, enter your conflict of interest statement in the “Confidential to Editor” section, and submit your "Accept" recommendation.

Reviewer #1: All comments have been addressed

Reviewer #2: All comments have been addressed

2. Is the manuscript technically sound, and do the data support the conclusions?

Reviewer #1: Yes

Reviewer #2: Yes

3. Has the statistical analysis been performed appropriately and rigorously? 

Reviewer #1: Yes

Reviewer #2: Yes

4. Have the authors made all data underlying the findings in their manuscript fully available?

Reviewer #1: Yes

Reviewer #2: Yes

5. Is the manuscript presented in an intelligible fashion and written in standard English?

Reviewer #1: Yes

Reviewer #2: Yes

6. Review Comments to the Author

Reviewer #1: Authors addressed all the concerns of reviewers. No additional comments. Authors provided detailed and valid explanation reviewer comments.

Reviewer #2: (No Response)

7. PLOS authors have the option to publish the peer review history of their article (what does this mean?). If published, this will include your full peer review and any attached files.

Reviewer #1: **Yes: **Chinthalapally V Rao

Reviewer #2: No

---

## [Editor Report · Acceptance letter]

22 May 2024

PONE-D-24-01608R1 

PLOS ONE

Dear Dr. Vázquez-Cuevas, 

I'm pleased to inform you that your manuscript has been deemed suitable for publication in PLOS ONE. Congratulations! Your manuscript is now being handed over to our production team.

Kind regards, 

on behalf of

Dr. Altaf Mohammed 

Academic Editor

PLOS ONE